

# Comparison of surface mass balance of ice sheets simulated by positive-degree-day method and energy balance approach

Eva Bauer, Andrey Ganopolski

Potsdam Institute for Climate Impact Research, Potsdam, Germany

*Correspondence to:* Eva Bauer (eva.bauer@pik-potsdam.de)

**Abstract.** Glacial cycles of the late Quaternary are shaped by the asymmetrically varying mass balance of continental ice sheets in the Northern Hemisphere. The surface mass balance is mostly positive during about four precssional periods and turns strongly negative at glacial terminations. The surface mass balance is governed by processes of ablation and accumulation. Here two ablation schemes, namely the positive-degree-day (PDD) method and the surface energy balance (SEB) approach, are compared in transient simulations of the last glacial cycle with an Earth system model of intermediate complexity. The standard version of the CLIMBER-2 model simulates ice volume variations reasonably close to reconstructions. It uses the SEB approach which comprises fluxes of short-wave and long-wave radiation and of sensible and latent heat and accounts explicitly for snow albedo changes from dust deposition and snow aging. The PDD-driven ablation is computed offline in ensemble simulations to study the sensitivity with respect to short-term temperature variability and to melt factors for snow and ice. With standard literature values, the anomaly between the 130 ka-long ablation series from the two schemes is minimized but, more suitable are smaller values for inception than for termination and larger values for ice sheets in America than in Europe. Accordingly, PDD-online simulations require smaller values for inception than for termination to reproduce global ice volume variations. However, a reproduction at inception involves afterward excessive ice volume growth up to twice as large as reconstructed at LGM while a reproduction at termination implies ice volume growth about half as reconstructed at LGM. The PDD-online simulation with standard values generates at LGM a huge sea level drop of 250 m and a global cooling of 8 °C. The PDD-online simulation reproducing the LGM ice volume produces insufficient ablation at the turning point from glacial to interglacial climate, hence termination is delayed. According to our simulations, the SEB approach including effects of changing snow albedo, in particular at the American ice sheet margins, proves superior for simulating glacial cycles.

## 1 Introduction

Glacial-interglacial cycles of the Quaternary are characterized by massive gains and losses of continental ice mass. The difference between gains from processes of snow accumulation and losses from processes of surface ice ablation defines the surface mass balance of ice sheets. The net surface mass balance is the volumetric change across the entire ice sheet and across a full accumulation and melt season constituting a balance year. The surface mass balance obtained from local measurements of the amounts of snow accumulated in winter and of snow and ice melted in summer is given in $\mathrm{mm\,water\,equivalent\,per\,year}$ which is usually shortened by $\mathrm{mm\,y^{-1}}$. The total mass balance of ice sheets includes further dynamic mass losses through





calving and basal melt but the surface mass balance is the main factor affecting the evolution of ice sheets on long (e.g. orbital) time scales.

Surface accumulation is connected to climate change through the hydrological cycle and results mostly from snowfall. Surface ablation is controlled by the surface energy balance (SEB) which primarily depends on air temperature and insolation.

During inception of a glacial cycle, accumulation predominates and ice sheets build up while at glacial termination the ablation predominates and ice sheets retreat. Numerical modeling shows that accumulation and ablation of the major ice sheets in America and Europe are very close to each other for most of glacial time period (Ganopolski et al., 2010). Hence, to simulate successfully the relatively slow buildup and the relatively rapid retreat of ice sheets during a glacial cycle depends crucially on an adequate description of accumulation and ablation processes.

Two alternative methods are in use to simulate ablation. One method is the so-called positive-degree-day (PDD) method. This is a semi-empirical parameterization which requires information only about surface air temperature (usually, monthly mean values). This method is computationally fast and therefore widely used to simulate the surface mass balance of ice sheets both in past (Tarasov and Peltier, 1999, 2002; Zweck and Huybrechts, 2003, 2005; Charbit et al., 2007; Abe-Ouchi et al., 2007; Lunt et al., 2008; Gregoire et al., 2012; Beghin et al., 2014; Liakka et al., 2016) and in future climate simulations (van de

Wal and Oerlemans, 1997; Huybrechts and de Wolde, 1999; Greve, 2000; Huybrechts et al., 2004; Ridley et al., 2005; Charbit et al., 2008; Winkelmann et al., 2015). The PDD method can be calibrated by use of measurements from glacier's surfaces but different glaciers give different values for the PDD scaling parameters. The other method is the physically-based SEB method which computes the energy available for melting of snow or ice in case the ice sheet surface temperature is above melting point (T > 273.15 K). This method requires calculations of all components of the energy balance (short-wave and long-wave

radiation, sensible and latent heat fluxes) which, in turn, requires a complete set of meteorological conditions. This method is computationally much more demanding than the PDD method and therefore was used till recently mostly in the framework of regional climate models for short-term climate predictions (Bougamont et al., 2006; Box et al., 2006, 2012; Fettweis, 2007, 2013; Ettema et al., 2009). However, simulations with a comprehensive Earth system model demonstrated that feedback effects between climate and ice sheets play an important role for simulating the ice mass balance in future climate change scenarios

(Vizcaino et al., 2010).

In spite of the obvious advantages of the PDD method for modeling the long-term climate-ice sheet interaction, such as Quaternary glacial cycles, there is a growing body of evidence that the PDD method is inadequate to simulate glacial cycles. One obvious problem with the PDD method is that it does not explicitly account for the absorption of short-wave radiation which represents the main energy component of the SEB. This can lead to significant underestimation of the effect from the

varying insolation on orbital time scales which is seen the primary driver of the glacial cycles (Robinson et al., 2010; van de Berg et al., 2011). Also for the reason of ignoring the short-wave absorption, the PDD approach cannot account explicitly for impacts of impurities (dust and soot) on the surface mass balance of the ice sheet. This could be a serious problem since paleoclimate data indicate significant increases of eolian dust deposition during glacial times, especially along the southern margins of the Northern Hemisphere (NH) ice sheets (Kohfeld and Harrison, 2001; Mahowald et al., 2006). Both theoretical

analysis (Warren and Wiscombe, 1980; Aoki et al., 2011) and direct measurements (Painter et al., 2010, 2012; Skiles et al.,





2012; Bryant et al., 2013; Doherty et al., 2013, 2014; Gautam et al., 2013) demonstrate that even a small amount of impurities affects the surface albedo significantly. In turn, results from SEB simulations show that these changes in albedo might significantly affect the surface mass balance of ice sheets during glacial times (Krinner et al., 2006; Ganopolski et al., 2010) and in future scenarios (Dumont et al., 2014; Goelles et al., 2015). At last, numerical parameters for the PDD method can only

be derived from observations over the existing ice sheets, primarily Greenland, and it is unclear a priory how different such parameters should be when the PDD method is applied to completely different climate conditions and different geographical distributions of ice sheets during glacial times.

So far, a direct comparison between PDD modeling and SEB modeling in a coupled climate-ice sheet model over a glacial cycle is missing. Here we discuss results from ensemble simulations of more than hundred transient simulations for the last

glacial cycle. In contrast to the study of Charbit et al. (2013) on different parameterizations inserted in a PDD model, we investigate space-time differences in the surface mass balance from using either the PDD method or the SEB approach. We use an Earth system model of intermediate complexity (EMIC) which simulates the last glacial cycle in a computationally efficient manner, though with the side effect that the description of details of the physical processes is limited. The EMIC consists of the climate model CLIMBER-2 interactively coupled with the ice sheet model SICOPOLIS. The standard version of the model

uses the SEB approach and the simulated ice volume changes, expressed by sea level variation, agrees reasonable with the sea level reconstructions of Waelbroeck et al. (2002). We take the space-time evolution simulated by the model as reference and compute in parallel the ablation by the PDD method. Thus, horizontally resolved ablation rates from two different methods can be compared under identical environmental conditions during the entire glacial cycle. In a second set of simulations, we exchange the SEB-derived ablation for the PDD-derived ablation and evaluate the glacial cycle simulations using the PDD

method in online mode against sea level reconstructions.

## 2   Model description

### 2.1   Model setup

The setup of the climate model CLIMBER-2 coupled with the ice sheet model SICOPOLIS is used as in Calov et al. (2005) and Ganopolski et al. (2010). CLIMBER-2 consists of interactively coupled models for the atmosphere, the ocean and the

vegetation. The atmospheric fields are computed on a longitude × latitude grid containing 7×18 grid cells. SICOPOLIS operates on the NH between 21 and 85.5 °N on a longitude, latitude grid $(x_s, y_s)$ with a resolution of $(1.5°, 0.75°)$. Thus one CLIMBER-2 grid cell can overlap with more than 450 grid cells of SICOPOLIS. CLIMBER-2 computes the atmospheric fields with a daily time step, the oceanic fields every five days and the vegetation distribution every year. SICOPOLIS computes the ice sheet evolution over a balance year. The CLIMBER-2 and SICOPOLIS models are coupled once per 10 years through

the interface module SEMI (Surface Energy and Mass balance Interface) which works on the fine SICOPOLIS grid. SEMI computes the surface mass balance and the surface temperature with a 3-day time step and transfers the annual fields of surface ice mass balance and of surface temperature to the SICOPOLIS model. In turn, SICOPOLIS feeds back to CLIMBER-2 the average ice sheet elevation, the fraction of land area covered by ice sheets, the sea level and the freshwater flux into the ocean



from the ablation of ice sheets and from ice calving. Further processes as avalanches and windblown snow are not considered and ice streams, meltwater channels or ice-dammed lakes are not resolved. This model configuration has been used before in glacial cycle simulations to study the 100 ka climatic cyclicity of the Quaternary (Ganopolski and Calov, 2011), the mineral dust cycle (Bauer and Ganopolski, 2010), the climate response to the dust radiative forcing (Bauer and Ganopolski, 2014) and
the impact of permafrost (Willeit and Ganopolski, 2015).

## 2.2   Surface energy and mass balance interface (SEMI)

The interface module SEMI solves the prognostic equations for ice surface temperature and snow thickness based on SEB and computes the surface mass balance on the fine grid of the SICOPOLIS model (Calov et al., 2005). The SEB comprises short-wave and long-wave radiative fluxes and turbulent energy fluxes and utilizes information from the CLIMBER-2 model
on insolation, temperature at the surface and in the near-surface air, wind and ice sheet elevation. The surface ablation is computed from a surplus in $SEB$ values and is hereafter called SEB-derived ablation. The ablation depends on snow layer thickness and snow albedo which is a function of dust deposition from aeolian and glaciogenic sources and snow aging. Then, SEMI computes the surface mass balance $F_{SEB}(x_s, y_s)$

$$F_{SEB}(x_s, y_s) = P(x_s, y_s) - A_{SEB}(x_s, y_s) \tag{1}$$

where $A_{SEB}(x_s, y_s)$ is the SEB-derived ablation (positively defined) and $P(x_s, y_s)$ is the snow accumulation. The determination of $P$ includes the elevation-desert effect which causes decreasing $P$ with increasing ice sheet elevation, and the elevation-slope effect which causes increasing $P$ with increasing slope of the ice sheet surface in up-wind conditions (Calov et al., 2005). Sublimation is here neglected.

## 2.3   Positive-degree-day (PDD) method

The PDD-derived ablation is calculated for a balance year on the SICOPOLIS grid inside the SEMI module. The PDD method is based on the reasoning that ablation is driven by the sum of positive daily temperature values which is seen as a proxy for melt energy (Braithwaite, 1984; Braithwaite and Olsen, 1989; Reeh, 1991). The semi-empirical PDD method represents a linear relation between the $PDD$ value and uses proportionality factors for snow and ice melt. Values of the melt factors which would be suitable for buildup and retreat of ice sheets over the entire glacial cycle are not known (Hock, 2003). In the
following, potential values of the melt factors are explored by ensemble simulations first in offline mode and second in online mode.

The $PDD$ value (in °C d) is defined as excess of daily surface air temperature above the melting point accumulated over a balance year. Because most implementations of the PDD method take daily temperature values from interpolated monthly mean climatological data and to account for the missing diurnal cycle and synoptic variability, a temperature variability term
is added. The short-term temperature variability may implicate melt occurrences, in particular at ice sheet margins, even if the mean temperature is negative. Usually, the standard deviation for temperature ($\sigma$) is prescribed in the range 4.5-5.5 °C (Reeh,




1991; Ritz et al., 1997; Tarasov and Peltier, 1999, 2002; Greve, 2005). Fausto et al. (2009) analyzed observations and showed that $\sigma$ for the Greenland ice sheet may increase from 1.6 to 5.2 °C for altitude increasing from 0 up to 3000 m.

The $PDD$ value is computed from the integral over time $t$

$$PDD = \int_{\Delta t} dt \, [\frac{\sigma}{\sqrt{2\pi}} exp(-\frac{T^2}{2\sigma^2}) + \frac{T}{2} \, erfc(-\frac{T}{\sqrt{2}\,\sigma})] \tag{2}$$

where $\Delta t = 1$ year, $T$ (in °C) is the 3-day mean of surface air temperature, $erfc(x)$ is the complementary error function and $\sigma$ is the standard deviation for temperature (Calov and Greve, 2005). The PDD-derived ablation is defined analogous to Eq. (1)

$$A_{PDD}(x_s, y_s) = P(x_s, y_s) - F_{PDD}(x_s, y_s) \tag{3}$$

where $P(x_s, y_s)$ in Eq. (1) and Eq. (3) are computed in the same way in SEMI. In the offline simulations, $P(x_s, y_s)$ in Eq. (1) and Eq. (3) are identical because they result from the environmental conditions of the reference simulation. The surface mass

balance $F_{PDD}$ (in mm y$^{-1}$) is calculated by

$$F_{PDD} = \begin{cases} \alpha_I \, Q & : \quad Q < 0 \\ 0 & : \quad Q = 0 \\ \alpha_S \, (1 - r_S) \, Q & : \quad Q > 0 \end{cases} \tag{4}$$

where $\alpha_S$ and $\alpha_I$ (in mm °C$^{-1}$ d$^{-1}$) are the melt factors of snow and ice, respectively, and $r_S = 0.3$ is a constant refreezing factor. This factor is introduced for the nocturnal refreezing of snows and causes a slow down of the snow melt. The factor $Q$ (in °C d y$^{-1}$) is the actual remain of $PDD$ per year $\Delta t$

$$Q = \frac{PDD_S - PDD}{\Delta t} \tag{5}$$

where $PDD$ is given in Eq. (2) and $PDD_S$ is

$$PDD_S = \frac{P\,\Delta t}{\alpha_S \, (1 - r_S)} \tag{6}$$

which represents that $PDD$ value which is supposedly required to melt the annual accumulated snow $P$. The sign of $Q$ determines the sign of the surface mass balance $F_{PDD}$. When the $PDD$ value (Eq. 2) is too small to melt the available snow

then the remaining snow at the end of the balance year builds ice mass and $F_{PDD}$ is positive. Reversely, when the $PDD$ value is large enough to melt all snow in the grid cell then the remain $Q$ (Eq. 5) melts surface ice and $F_{PDD}$ is negative.

## 2.4   Reference simulation of last glacial cycle

The reference simulation of the last glacial cycle utilizes the SEB-derived ablation from SEMI. All glacial cycle simulations are driven by the insolation calculated from the varying orbital parameters (Berger, 1978) and the varying concentration of

greenhouse gases (Fig. 1a) expressed as equivalent $CO_2$ concentration (Ganopolski et al., 2010). The radiative forcing by aeolian dust and the aeolian dust deposition on snow affecting the snow albedo of ice sheets are computed by using time





slice simulations from general circulation models. Temporally varying fields are obtained by scaling the time slices with the simulated ice volume (Ganopolski et al., 2010). The snow albedo of the ice sheets depends further on dust deposition from internally simulated glaciogenic dust sources and snow aging.

Figure 1b shows the reference time series of global mean surface air temperature and global mean precipitation over $130\,\mathrm{ka}$.

The global temperature $T$ decreases irregularly by more than $6\,^{\circ}\mathrm{C}$ from the last interglacial, the EEM, until $21\,\mathrm{ka}$, the last glacial maximum (LGM). Subsequently, $T$ rises rapidly by $5.5\,^{\circ}\mathrm{C}$ within about $10\,\mathrm{ka}$ until the early Holocene. The global precipitation is thermodynamically controlled and varies in close relationship to $T$ (Fig. 1b). Figure 1c shows the mean sea level variation computed from the NH ice volume (assuming constant ocean surface area and an additional 10% contribution from the Antarctic ice sheet) in comparison to the global mean sea level from reconstructions (Waelbroeck et al., 2002).

Figure 2 shows the characteristics of the NH ice sheets by comparing NH total values with values from two main partitions. Hereafter, we name the two partitions the American and the European ice sheets which represent, respectively, the ice sheets on the northern American continent and in northwestern Eurasia extending as far as $120\,^{\circ}\mathrm{E}$. Up to $70\,\%$ of the total ice-covered area occurs on the American continent and mostly less than $20\,\%$ occurs in Europe (Fig. 2a). The total ice volume varies about proportional to the total ice sheet area (Fig. 2b). The volume of the ice sheets is given in $\mathrm{meter\,sea\,level\,equivalent\,(msle)}$

which is determined under the assumption that changes in the ocean surface area are negligibly small over the glacial cycle. The area and the volume vary inversely proportional to the precession-driven variation of the northern summer insolation. The local maxima due to the precessional varying insolation grow gradually until LGM where the growth of ice volume is more steep than the growth of ice area.

Figure 2c shows areal averages of ice sheet thickness. These have to be interpreted with care because the area-weighted

average thickness is related to the variable ice sheet area. In the glacial period, the American ice sheet thickness is larger than the total average thickness whereas the American ice area is a fraction (although major) of the total ice-covered area. During the interglacial periods, the relatively high average thickness of the total ice sheet is related to the persisting Greenland ice sheet. In the initial millennia of glacial inception, the drop in the average ice thickness results from the fast spreading of the ice sheet area during inception (Calov et al., 2005). Thereafter the average thickness of the American ice sheet grows, stays high

beyond the LGM and drops rapidly toward to beginning of the Holocene. The European ice sheet thickness starts to grow at glacial inception a few millennia before the American ice thickness. The European average ice thickness shows relatively little variations in the glacial four precessional periods while the area and volume vary with precessional periods. Around the LGM, the European ice thickness increases by about $30\,\%$ which is accompanied with an extra cooling over the northern Atlantic. The lead of the thinning of the European ice sheet compared to American ice sheet at glacial termination is attributed to the

lower elevation of the European ice sheet which facilitates the ice melt. Yet, the thinning of the American ice sheet occurs more rapidly than of the European ice sheet.

The components of the surface mass balance are shown in Sv ($1\,\mathrm{Sv} = 10^6\,\mathrm{m}^3\,\mathrm{s}^{-1}$) for the total ice sheet together with the partitions from America and Europe. The snow accumulation (Fig. 2d) is well correlated with the ice sheet area and varies with the precessional period in a rather harmonic manner. The ablation (Fig. 2e) varies irregularly in response to different

driving factors, as insolation, surface ice area exposed to temperature above melting point and albedo of the snow surface. The



maximum ablation after the LGM occurs in America some millennia earlier than in Europe. The lead of the maximum ablation in America is related to the larger perimeter exposed to melt conditions and the more southerly extent of the American ice sheet. The resulting surface mass balance (Fig. 2f) is positive during the glacial period leading to the buildup of ice mass. After the LGM the mass balance turns negative and the ice sheets retreat. In the time interval around 65 ka (MIS 4) the simulated

negative surface mass balance results from a climatic excursion involving interactions between atmospheric cooling amplified by dust short-wave radiative forcing and changes in the North Atlantic meridional overturning circulation (AMOC).

## 3  Mass balance by offline PDD method during glacial cycle

Any simulation over the last glacial cycle with the PDD method suffers from missing empirical data which are necessary to calibrate the PDD method. Therefore, we use the reference simulation for the last glacial cycle and compute ensembles of

PDD-derived ablation in offline mode by varying the parameter values. The standard deviation for temperature $\sigma$ (Eq. 2) and melt factors $\alpha_S$ and $\alpha_I$ (Eq. 4) are considered as control parameters. Each simulation is run with constant parameter values.

### 3.1  Selection of PDD parameter values

The $PDD$ value (Eq. 2) is calculated with prescribed standard deviation for temperature. We insert two different values, i.e., $\sigma$=3 °C and $\sigma$=5 °C. Figure 3 shows time series of $T$ and the corresponding $PDD$ values as areal averages over the ice sheets.

The temperature averaged over the NH ice sheet area decreases by 13 °C (from -16 to -29 °C) after the last interglacial in a time interval of about 100 ka and then $T$ returns rapidly within about 10 ka, thus shaping the asymmetry of the glacial cycle (Fig. 3a). The $PDD$ values are closely correlated with $T$ showing a progressive decrease after glacial inception and a rapid increase during glacial termination. The areal averages of the $PDD$ value lie in the range 10–70 °C d with $\sigma$=3 °C and lie in the range 20–120 °C d with $\sigma$=5 °C (Fig. 3a). The asymmetric evolution is substantiated mostly by the temperature evolution

over the massive ice sheet on the American continent (Fig. 3b) ranging from -16 to -27 °C. The temperature of the European ice sheet fluctuates strongly between -10 and -29 °C. These fluctuations are connected with changes in the sea-ice albedo effect in the northern Atlantic and changes in the heat advection by the AMOC. The European $PDD$ values range over 10–260 °C d with $\sigma$=3 °C and range over 30–370 °C d with $\sigma$=5 °C (Fig. 3c).

Previous climate model studies often used $\sigma$ about 5 °C and standard melt factors, i.e., $(\alpha_S, \alpha_I) = (3, 8)\,\mathrm{mm\,°C^{-1}\,d^{-1}}$

which were derived from measurements on the Greenland ice sheet (Huybrechts and de Wolde, 1999; Tarasov and Peltier, 1999, 2000). However, observations show that melt factors may vary with latitude and height of the glacier. Worldwide measurements during the melt season of glaciers and snow-covered basins yield melt factors for $\alpha_S$ and $\alpha_I$ (in $\mathrm{mm\,°C^{-1}\,d^{-1}}$) in the ranges [2.5–11.6] and [5.4–20], respectively (Hock, 2003). We study the PDD-derived ablation by varying $\alpha_S$ and $\alpha_I$ (in $\mathrm{mm\,°C^{-1}\,d^{-1}}$) in case $\sigma$=3 °C in the ranges [3-10] and [8-24], respectively, and in case $\sigma$=5 °C in the ranges [2-6] and

[4-18], respectively, Thereby the entire variability of the SEB-derived ablation simulated for the ice sheets is enclosed by the ensembles of the offline PDD-derived ablation which is shown in Fig. 6 for the American and the European ice sheets.



### 3.2 Ablation time series for American and European ice sheets

We use as measures of agreement between the reference and the PDD method the mean anomaly $m$ and the rms–error $r$ averaged over the NH ice sheet (or over the American and European ice sheets) and the entire glacial cycle (or shorter time intervals). Figure 4 compares ablation series from the PDD method with the reference series over the last 130 ka averaged for the American and European ice sheets. The five ensemble members for each $\sigma$-set which are selected for Fig. 4 show relatively small mean anomalies (see Tab. 1 for corresponding $m$ and $r$). The ensemble members with $\sigma$=3 °C (Fig 4a, b) are produced with $\alpha_S = 5\,\mathrm{mm\,°C^{-1}\,d^{-1}}$ and $\alpha_I$ in the range 8–24 $\mathrm{mm\,°C^{-1}\,d^{-1}}$, and the ensemble members with $\sigma$=5 °C (Fig 4c, d) are produced with smaller melt factors, i.e., $\alpha_S = 3\,\mathrm{mm\,°C^{-1}\,d^{-1}}$ and $\alpha_I$ in the range 4–12 $\mathrm{mm\,°C^{-1}\,d^{-1}}$. For the American ice sheet, $m$ is between -0.023 and -0.007 Sv and for the European ice sheet, $m$ is between -0.011 and 0.006 Sv in case $\sigma$=3 °C. In case $\sigma$=5 °C, the minimum anomaly $m$ for both ice sheets is found by use of the standard melt factors, i.e. $(\alpha_S, \alpha_I)$ = (3, 8) $\mathrm{mm\,°C^{-1}\,d^{-1}}$. Apparently, the agreement between the series is much lower for the American than for the European ice sheets irrespective of the $\sigma$ value. An outstanding feature is the enhanced ablation from the American ice sheet during MIS 4 which is difficult to reproduce with the PDD method.

Another selection of ensemble simulations looks for minima in rms–error. Minima in $m$ and minima in $r$ do not necessarily have common melt factors (Tab. 1). The contour plots of the rms–error shown as a function of $\alpha_S$ and $\alpha_I$ illustrate that no unique pair $(\alpha_S, \alpha_I)$ is suitable for both ice sheets (Fig. 5). Overall, $r$ for the American ice sheet is about a factor three larger than for the European ice sheet in both $\sigma$-sets.

The PDD-derived ablation series produced with the smallest and largest $\alpha_S$ and $\alpha_I$ values envelop the reference ablation series for the American and the European ice sheets (Fig. 6). As seen already in Fig. 4, a particular PDD-derived ablation series which overestimates the SEB-derived ablation during glacial inception underestimates the peaks of the reference ablation at glacial termination. Hence, optimal melt factors for glacial inception are most likely smaller than for glacial termination. Table 2 gives an example for PDD parameter values which produce minimum rms–errors for shorter time intervals, first for 130–30 ka and second for the last 30 ka. Nonetheless, ablation series fitted separately for the American ice sheet deviate repeatedly from the irregularly fluctuating reference series (Fig. 6a, c). The PDD-derived ablation series fitted for the European ice sheet for these time intervals, however, agree quite well with the reference and the discontinuity at 30 ka is small (Fig. 6b, d).

### 3.3 Ablation rates on fine resolution at glacial termination

At 15 ka, the total SEB-derived ablation has a maximum of 0.41 Sv (Tab. 3). The ensemble member which produces about the same total ablation as the reference at 15 ka is obtained with $\sigma$=3 °C and $(\alpha_S, \alpha_I)$=(9, 16) $\mathrm{mm\,°C^{-1}\,d^{-1}}$ (Fig. 7). But that ensemble member produces at 15 ka a smaller American ice melt and a larger European ice melt than the respective references (Fig. 7). However, that ensemble member produces for the European ice sheet a maximum in ablation at 14 ka which is also seen in the reference (Tab. 3). No single ensemble simulation is found that can produce for both ice sheets in America and in Europe ablation maxima at the same time instances as the reference simulation.



The comparison of the ablation rates of both methods on the fine SICOPOLIS grid shows, in case of equal NH total ablation, that the PDD method tends to overestimate large ablation rates and to underestimate low ablation rates. This is demonstrated in a scatter diagram (Fig. 8) comparing the ablation rates at 15 ka from the above ensemble simulation (Fig. 7) with the reference. The PDD-derived American melt rates overestimate the reference ablation rates larger than $\sim 10 \, \mathrm{mm \, d^{-1}}$ but underestimate

the American ice melt rates less than $\sim 8 \, \mathrm{mm \, d^{-1}}$ (Fig. 8a). The PDD-derived European melt rates are overestimated mainly for ablation rates larger than $\sim 6 \, \mathrm{mm \, d^{-1}}$ (Fig. 8b). The largest ablation rates occur naturally at the ice sheet margins and here the largest differences are located. This can be seen in Fig. 9 from the geographic distribution of the differences between the PDD-derived ablation relative to the SEB-derived ablation at 15 ka. The differences are positive mostly at the outer margins of the ice sheets. Negative differences occur predominately around the Rocky Mountains.

**4   Glacial cycle simulations with online PDD method**

The PDD-derived ablation from the above offline simulations are evaluated against the SEB-derived ablation by assuming that the reference simulation provides acceptable climate characteristics since the reference simulation reproduces the reconstructed sea level reasonably well. In the following PDD-online simulations, the PDD-derived ablation replaces the SEB-derived ablation. In this way, the simulated climate and the ice sheets are internally consistent with the PDD method but thereby the

impact from changing snow albedo on the absorption of short-wave energy at ice sheet surfaces is ignored. We evaluate the PDD-online simulation by comparisons with the reconstructed sea level and climate characteristics of the reference simulation.

**4.1   Selection of PDD parameters values**

The globally and temporally constant PDD parameter values are selected with the aim to reproduce the reconstructed sea level at three target windows which are glacial inception, glacial termination and LGM. Table 4 lists the PDD parameter values

which are suitable for simulating the climate at the three target windows and which produce representative results. These representative results are obtained by applying the PDD method online in a set of glacial cycle simulations.

The targets at inception and termination could be fulfilled with a range of PDD melt factors and we select parameter values on the basis of the results from the offline simulations. The offline simulations of $A_{PDD}$ using $\sigma$=3 °C which produced minimum rms–errors for American and European ice sheets indicate that $A_{PDD}$ is mainly sensitive to the snow melt factor while the ice melt factor is invariant, namely $\alpha_I$=16 $\mathrm{mm \, °C^{-1} \, d^{-1}}$ (Tab. 2). So we use that $\alpha_I$ value and varied $\alpha_S$ to fulfill

the first two targets by PDD-online simulations. The offline simulations of $A_{PDD}$ with $\sigma$=5 °C yield differing melt factors at minimum rms–errors. We recall that with the standard PDD parameter values the mean anomaly in the PDD-offline simulations is minimized (Tab. 1). By use of the standard PDD parameter values in the PDD-online simulation the first target is fulfilled and the second target can be fulfilled by doubling the $\alpha_S$ value. The reproduction of the sea level at LGM emerged as a rather

strong constraint and only one specific pair of melt factors values for each $\sigma$ value is found suitable.



## 4.2 Target window: glacial inception and termination

The PDD-online simulations I3 and I5 (Tab. 4) reproduce closely the global temperature (Fig. 10a, c) and the sea level (Fig. 10b, d) during inception over the first precessional period. In this time interval the ice sheet area grows sufficiently fast in company with accumulation. The reproduction of $T$ implies reproductions of both the ice sheet thickness and the ablation and conse-

quently the surface mass balance agrees with the reference (not shown). Thereafter the ice volume grows too fast in concert with amplified snow accumulation. At 21 ka, the ice volume is about twice as large as reconstructed (Tab. 4) and then the simulations I3 and I5 fail to terminate the glacial climate state. Note, the simulation I5 which uses the standard PDD parameter values (Table 4) simulates the climate characteristics in the first multi-millennia in close agreement with the reference but then drifts into excessive cold climate without recurrence (Fig. 10c, d).

The temperature and the sea level in simulations T3 and T5 recover the Holocene climate characteristics after a weak glacial phase (Tab. 4). The global cooling after inception is about in phase with the reference temperature though the cooling in the PDD-online simulations is substantially underestimated (Fig. 10a, c). The sea level drop in simulation T3 is about half as large as reconstructed over the glacial phase (Fig. 10b) and in simulation T5, the maximum sea level drop is 40 m occuring after the LGM (Fig. 10d). From 38 to 20 ka the cooling rate in both simulations T3 and T5 intensifies and thereby the ice volume

grows continuously beyond 21 ka until around 18 ka. The recurrence to Holocene climate begins from a less cool climate and a smaller ice-covered area than in the reference. So, both simulations T3 and T5 undershoot the buildup of the ice volume substantially.

## 4.3 Target window: LGM

The PDD-online simulations L3 and L5 reproduce reasonably well the reconstructed sea level at 21 ka (Tab. 4). In the initial

phase of the glacial cycle, the simulation L3 produces a weaker cooling and less ice volume than the reference but in the time interval 40–21 ka the agreement is close (Fig. 10a, b). The simulation L5 with the high temperature variability generates a growing ice volume over the entire glacial phase which agrees well within uncertainties inferred from the reference and the reconstructed sea level (Fig. 10c, d). After the LGM, the ice volume in L3 and L5 grows further by several msle. The continued growth of the ice volume is associated with a continued positive mass balance from less ablation than in the reference simulation

mainly in America. Consequently, glacial termination is delayed and the recurrence to Holocene climate characteristics is not achieved.

The geographic distribution of the ice sheet thickness at 21 ka from the PDD-online simulation L3 agrees closely with the reference simulation (Fig. 11). The simulation L3 reproduces the maximum thickness of 3500 m in America as simulated by the reference. The simulation L3 produces a slightly more southerly spreading ice sheet beyond the American Great Lakes

and a thinner ice sheet in the European Arctic and in northeastern Asia. Also, simulation L5 produces an ice sheet distribution similar to the reference although the maximum thickness in America is only 3300 m at LGM. Both PDD-online simulations L3 and L5 simulate at LGM a sea level of -120 m, but thereafter their mass balances remain more positive than in the reference which results in lagged climate warming and in case of L5 the deglaciation is incomplete (ca. 50 msle remain at present).



## 5 Discussion

Simulations of the long-term climate changes during glacial cycles are here discussed with focus on the coupling mechanisms between the climate system and the ice sheet distribution. The coupling module SEMI between the CLIMBER-2 model and the relatively high-resolution SICOPOLIS model provides the ice sheet model with the surface ice mass balance and in turn

provides the climate model with the spatial distribution of the ice sheet. Differences in the surface mass balance computed by the PDD method and the SEB approach are studied in transient simulations over the last glacial cycle for the American and the European ice sheets and at a fixed output time for climate variables on the relatively fine geographic resolution.

The comparison of PDD-derived ablation and SEB-derived ablation accumulated for the ice sheets suggests that PDD melt factors should be larger for glacial termination than for glacial inception and larger for the American ice sheet than for the

European ice sheet. The rms–error between $A_{PDD}$ and $A_{SEB}$ for the American ice sheet is found threefold larger than for the European ice sheet (Tab. 1 and Fig. 5). Hence, the European ice sheet appears to be closer correlated with the positive temperature sum than the American ice sheet. The bivariate rms–error distributions show that low rms–errors in the ablation from the American ice sheet are more sensitive to the snow melt factor than to the ice melt factor, while the rms-error in ablation from the European ice sheet shows a similar sensitivity to both $\alpha_I$ and $\alpha_S$ (Fig. 5). Thus the simulated American ice

melt is seen to depend more closely on the snow melt factor which can be attributed to the effect of dust deposition on the American ice sheet.

Comparisons of the local ablation rates from the PDD and the SEB methods with climate variables on the fine SICOPO-LIS grid are blurred because of the large variability, for instance, with location, ice sheet thickness and absorbed short-wave insolation changing with snow albedo through dust deposition. Another reason for differing ablation rates is that the SEB

method includes influences from the nonlinear interplay of the short-term varying climate variables which are calculated with a 3-day time step. In order to reduce the deficiencies of the PDD method, already Braithwaite (1995) pointed out that melt factors should explicitly account for temperature, albedo and turbulence or in other words, better to employ the SEB approach. Therefore we analyze the PDD-derived ablation (offline) and the SEB-derived ablation for their relation to the concurrently simulated $SEB$ values. The largest differences between the ablation from the PDD and the SEB methods are visible at glacial

termination and we take the ablation rates simulated at $15\,\mathrm{ka}$ as used above (Fig. 7, 8, 9). Figure 12a clearly shows that $A_{SEB}$ grows steadily with $SEB$. The offline computed $A_{PDD}$ grows proportional with $PDD$ values and additionally $A_{PDD}$ grows slightly with $P$ (Fig. 12b). When $A_{PDD}$ is interpolated with respect to $SEB$ and $P$ then only the largest values of $A_{PDD}$ coincide with the largest $SEB$ values. The largest $A_{PDD}$ values are located at the outer margin of the American ice sheet and are seen to overshoot the corresponding $A_{SEB}$ values (Fig. 8). Clearly, most of the $A_{PDD}$ values (i.e., between 5 and

$15\,\mathrm{mm\,d^{-1}}$) vary randomly with $SEB$ (Fig. 12c). This poor correlation between $A_{PDD}$ and $SEB$ is an illustration of the shortcoming of the PDD method.





## 6 Conclusions

The overall target is to simulate the asymmetric evolution of the NH ice sheets of the last glacial cycle which build up over about 100 ka and retreat within about 10 ka. The changing surface ice mass balance plays a crucial role in shaping the glacial cycles of the Quaternary. The surface ice mass balance is the annual difference of accumulation minus ablation which have similar values

during most of the glacial period. Under the assumption that accumulation simulated by the CLIMBER-2 model is plausible since the asymmetric climate evolution is simulated satisfactorily, we compare the ablation from the offline PDD method and the SEB approach. The PDD-derived ablation is computed in a set of more than hundred transient glacial cycle simulations using in each simulation constant values for the temperature variability term and the PDD melt factors. The comparison between ablation series from the offline PDD method and the SEB method shows:

i) if the rms–error is small for the European ice sheet then $A_{PDD}$ is too low for the American ice sheet, and vice verse if the rms–error is small for the American ice sheet then $A_{PDD}$ is too large for the European ice sheet,

ii) if the rms–error is small at glacial inception then $A_{PDD}$ is too small at glacial termination, and vice verse if the rms–error is small at glacial termination then $A_{PDD}$ is too large at glacial inception.

This indicates that the PDD-derived ablation with constant parameter values is not compatible with the SEB-derived ablation

in long-term simulations with varying climate conditions and with geographically varying continental ice sheets as is observed in NH during glacial cycles.

The glacial cycle simulations using the PDD method in online mode can reproduce the reconstructed sea level quite well either for glacial inception or for glacial termination but only by use of different PDD parameter values for the different phases. Hence, those PDD-online simulations which generate a plausible sea level drop at inception overshoot strongly the sea level

drop at LGM and fail to recover the interglacial sea level. This is, for instance, true for the simulation using standard PDD parameter values. PDD-online simulations which recover the sea level of the Holocene produce prior, in the glacial phase, a rather flat sea level decrease and underestimate the sea level drop at LGM substantially. One triple of PDD parameters is found by which the sea level is simulated remarkable well during the glacial phase and at LGM. After the LGM, however, the ice volume grows further during several millennia by a few $\mathrm{meters\,sea\,level\,equivalent}$ and subsequently the sea level

rise at present is about $50\,\mathrm{msle}$ too low. No universal PDD parameter values are found by which the entire glacial cycle is simulated satisfactorily with the online PDD method. The simulations with the SEB approach suggest that the relatively fast spreading of the ice sheet area at inception and the snow albedo changes from dust deposition at termination are important elements for the glacial cycle evolution. This study motivates further investigations on the role of dust in the climate system. Influences of dust radiative forcing and dust deposition on snow of ice sheet surfaces are included in a simplified manner

in the current CLIMBER-2 simulations. Improvements are expected by using a dynamically and bio-geochemical consistent dust cycle model. The effects of interactions between the climate system and the dust cycle are seen to be variable during the Quaternary and are likely involved also in future climate change studies.

*Acknowledgements.* E.B. acknowledges support by the German Climate Modeling Initiative grant PalMod.



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





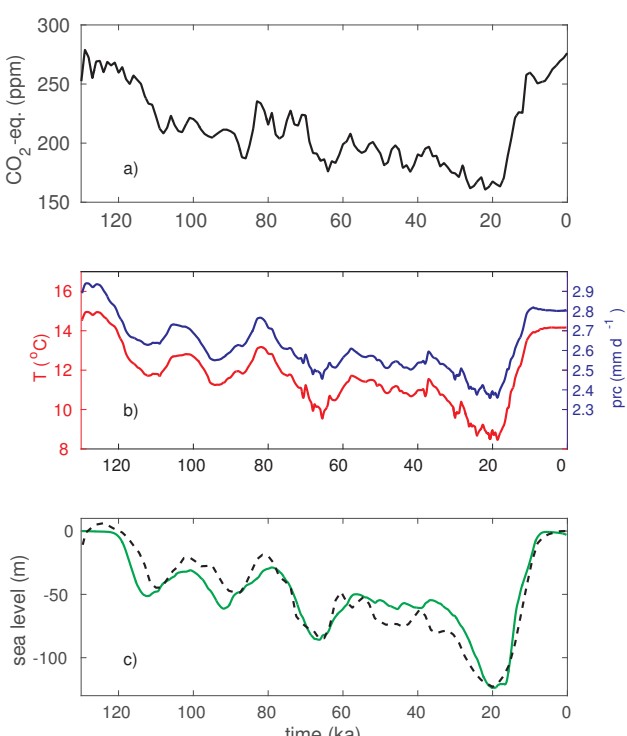

**Figure 1.** Reference simulation of last glacial cycle with CLIMBER-2 model coupled with SICOPOLIS model via SEB approach. **(a)** Driving equivalent $CO_2$ concentration, **(b, red)** global mean surface air temperature, **(b, blue)** global mean precipitation and **(c)** sea level shown by **green line** from simulated ice volume variation and by **black dashed line** from reconstructions by Waelbroeck et al. (2002).





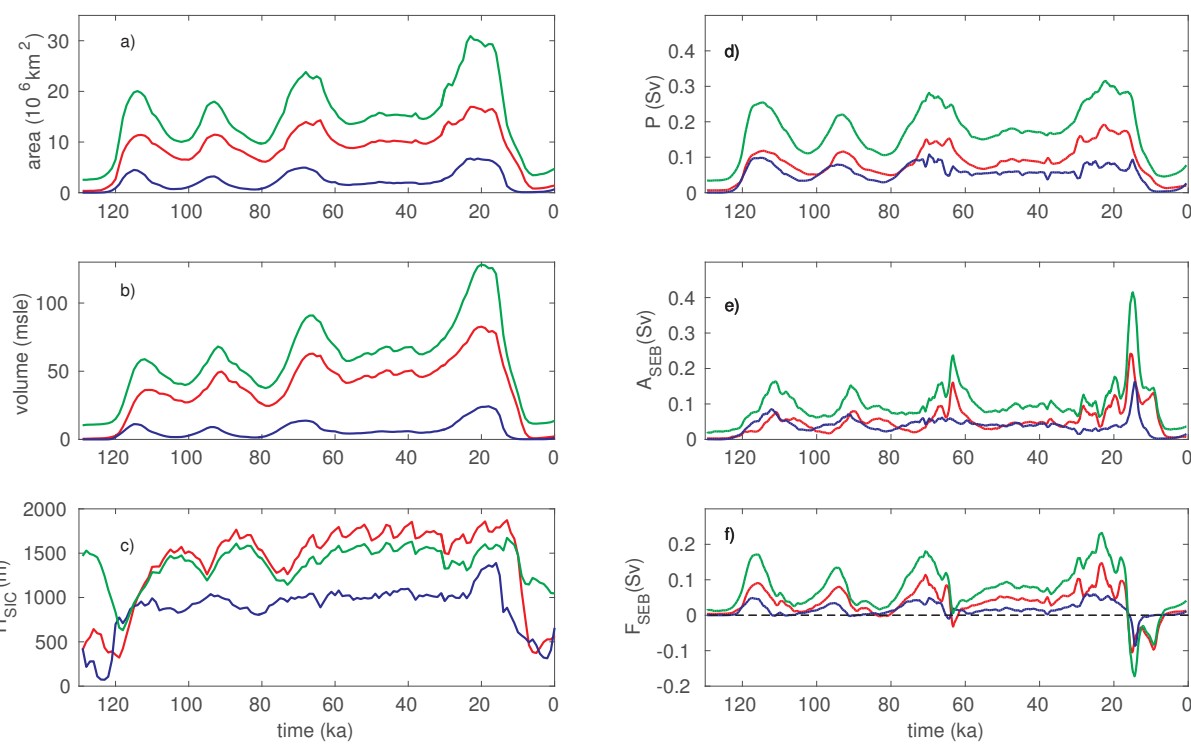

**Figure 2.** Glacial cycle series from reference simulation for NH total **(green lines)**, American **(red lines)** and European **(blue lines)** ice sheets showing **(a)** ice-covered area, **(b)** ice sheet volume, **(c)** average ice sheet thickness, **(d)** accumulation, **(e)** SEB-derived ablation and **(f)** surface ice mass balance.





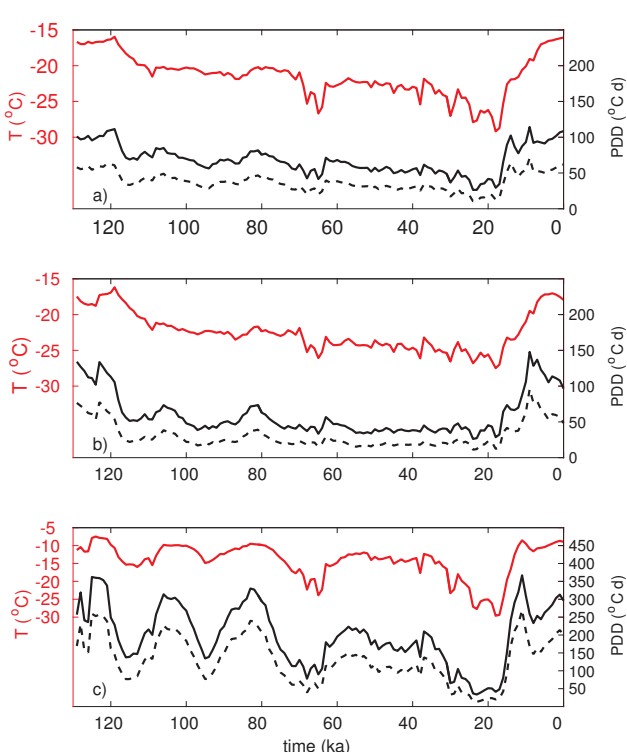

**Figure 3.** Glacial cycle series averaged over **(a)** NH total, **(b)** American and **(c)** European ice sheets showing on **left axes (red)** surface air temperature and on **right axes (black)** $PDD$ values (Eq. 2) computed with $\sigma$=3 $^\circ$C **(dashed lines)** and with $\sigma$=5 $^\circ$C **(continuous lines)**.



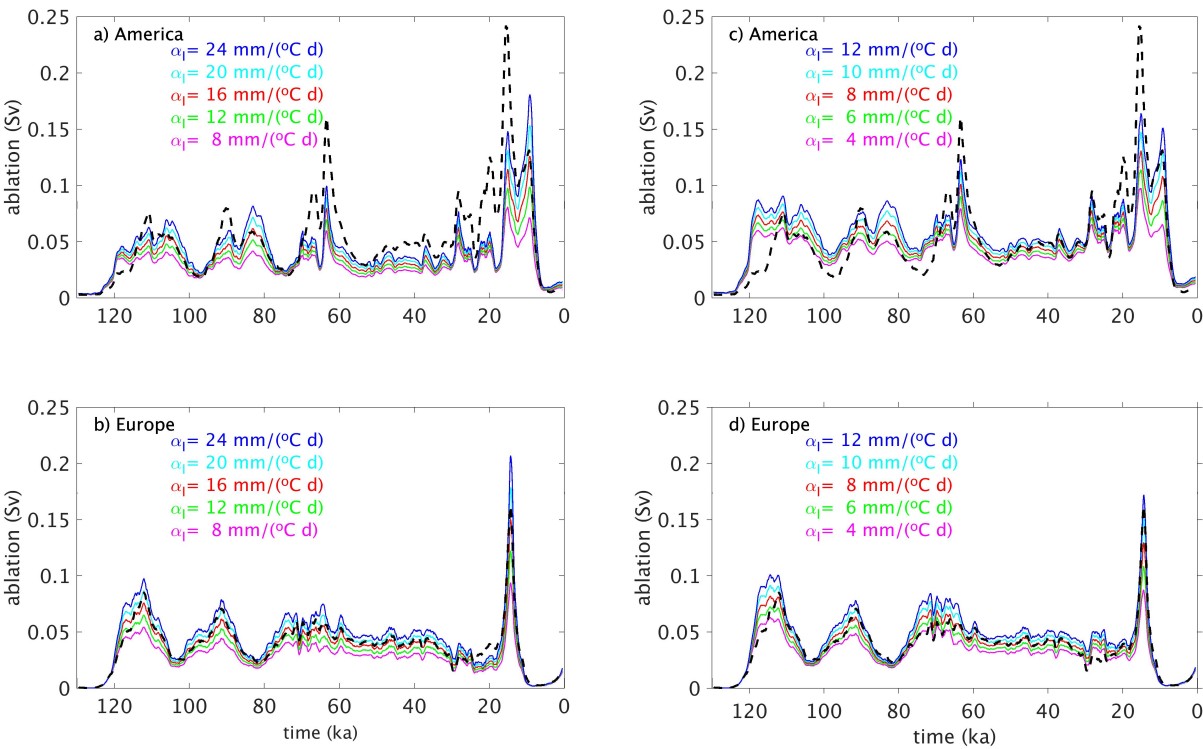

**Figure 4.** Glacial cycle series of ablation for **(a, c)** American and **(b, d)** European ice sheets comparing ensembles of offline PDD-derived ablation **(colored lines)** with SEB-derived ablation of reference simulation **(black dashed line)**. PDD-derived ablation use in **(a, b)** $\sigma$=3 °C, $\alpha_S = 5 \, \mathrm{mm} \, °\mathrm{C}^{-1} \, \mathrm{d}^{-1}$ and five different $\alpha_I$ values and in **(c, d)** $\sigma$=5 °C, $\alpha_S = 3 \, \mathrm{mm} \, °\mathrm{C}^{-1} \, \mathrm{d}^{-1}$ and five different $\alpha_I$ values. The different $\alpha_I$ values are shown by different colors in each panel. Note, **red lines** in (**c, d**) are obtained with standard PDD parameter values. See Tab. 1 for mean anomalies and rms–errors.





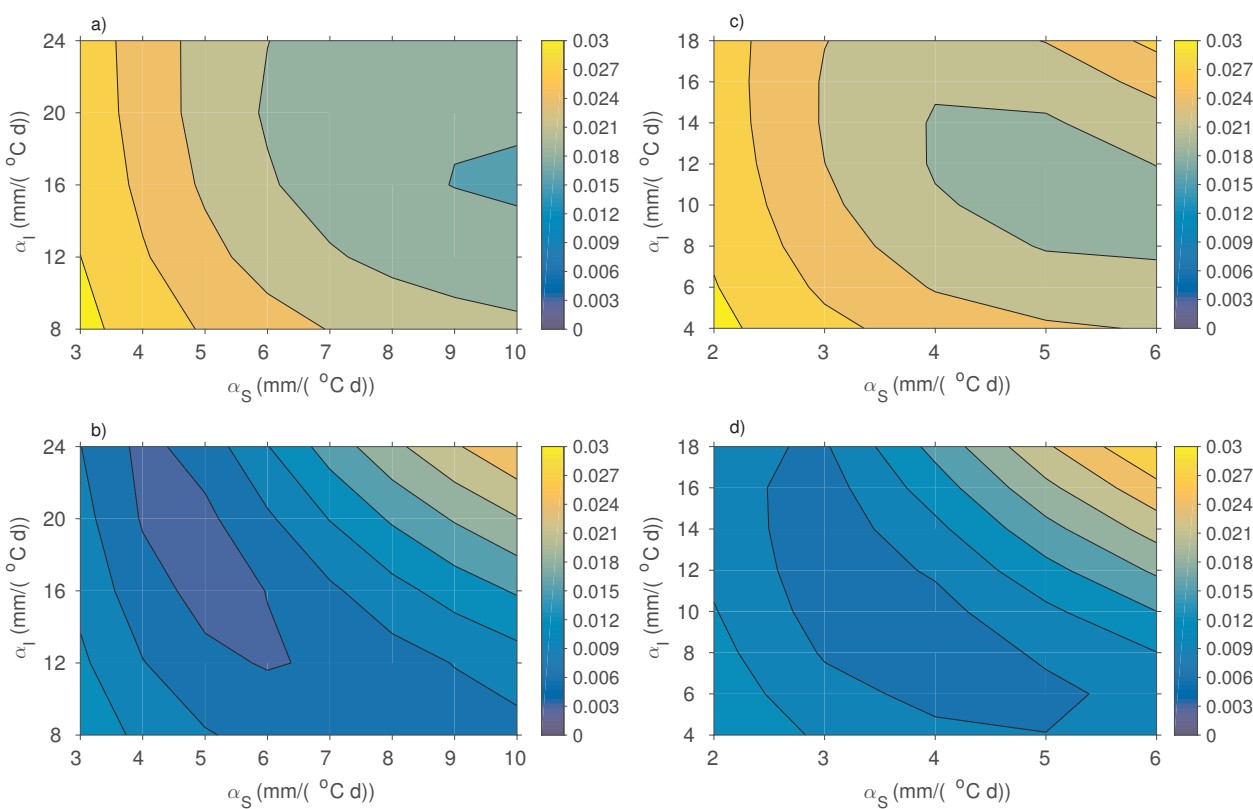

**Figure 5.** Bivariate distributions of rms–error $r$ (in Sv) as function of $\alpha_S$ and $\alpha_I$ where $r$ is from ensemble simulations of $A_{PDD}$ (offline) relative to $A_{SEB}$ using entire 130 ka-long series. Calculations of $r$ in **(a, c)** for American ice sheet and in **(b, d)** for European ice sheet. Ensemble simulations of $A_{PDD}$ use in **(a, b)** $\sigma$=3 °C and in **(c, d)** $\sigma$=5 °C which involves larger values for ($\alpha_S$, $\alpha_I$) in **(a, b)** than in **(c, d)**. See Tab. 2 for PDD parameter values at minimum of rms–error in each panel.





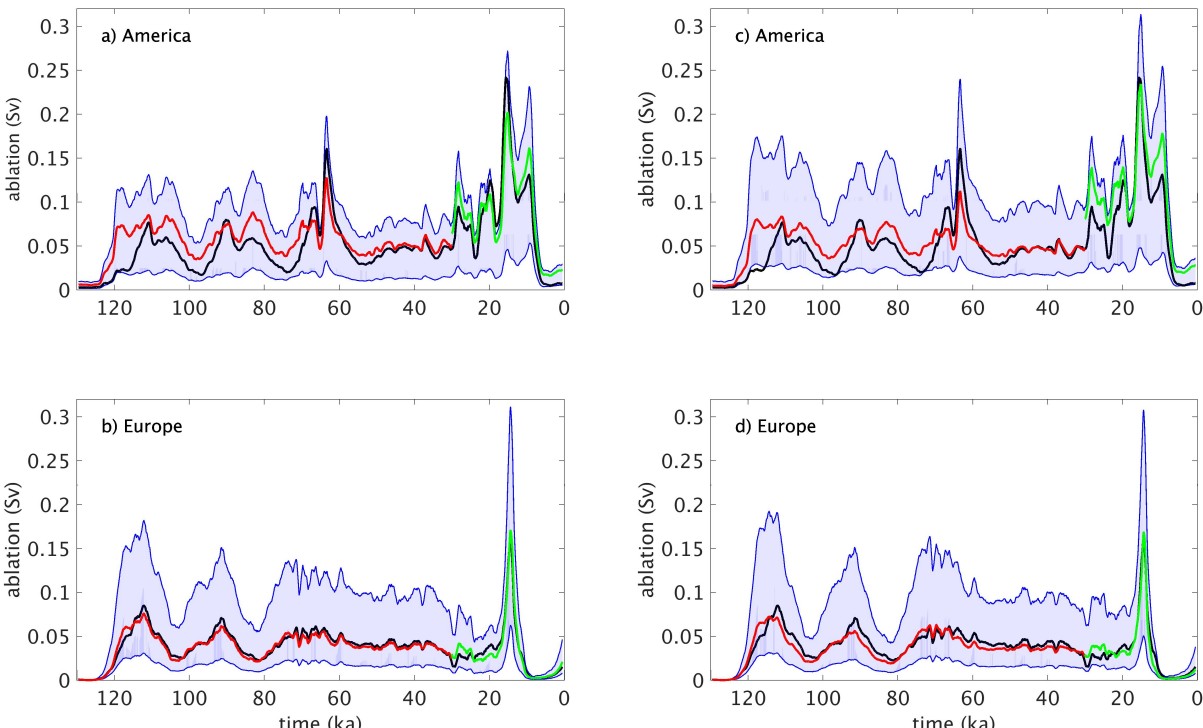

**Figure 6.** Glacial cycle series of ablation in **(a - d)** as in Fig. 4. **Black lines** are $A_{SEB}$ from reference simulation and **blue shaded areas** show full ranges of offline $A_{PDD}$ from ensemble simulations. PDD parameter values ($\sigma$, ($\alpha_S$, $\alpha_I$)) in (°C, (mm °C$^{-1}$ d$^{-1}$)) used for lower and upper boundary are in **(a, b)** (3, (3,8)) and (3, (10,24)), respectively, and in **(c, d)** (5, (2,4)) and (5, (6,18)), respectively. Further PDD-derived ablation series are shown by which rms–errors for American and European ice sheets minimize over 130–30 ka (**red lines**) and over 30–0 ka (**green lines**). PDD parameter values used in **(a)** in **red**: (3, (8,16)) and in **green**: (3, (10,16)), in **(b)** in **red**: (3, (5,16)) and in **green**: (3, (6,16)), in **(c)** in **red**: (5, (4,10)) and in **green**: (5, (6,12)) and in **(d)** in **red**: (5, (4,6)) and in **green**: (5, (3,16)). See Tab. 2 for summary of PDD parameter values at minima of $r$.





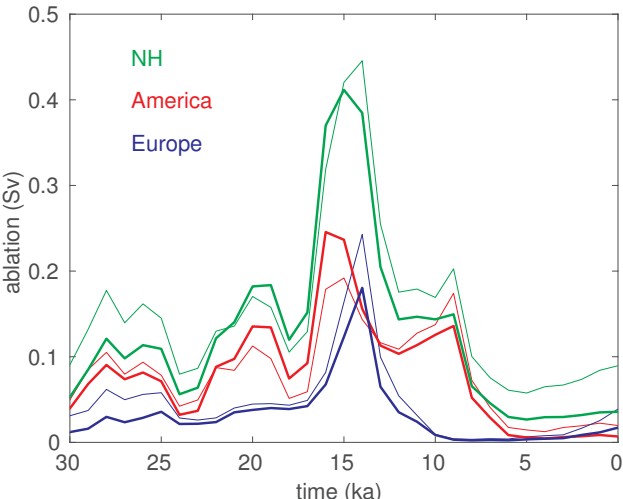

**Figure 7.** Ablation series from interval 30–0 ka for NH total (**green lines**), for American (**red lines**) and for European (**blue lines**) ice sheets showing $A_{SEB}$ of reference simulation by **thick lines** and offline $A_{PDD}$ by **thin lines**. $A_{PDD}$ with parameter values $\sigma = 3\,°C$ and $(\alpha_S\,\alpha_I) = (9, 16)\,mm\,°C^{-1}\,d^{-1}$ is compatible with $A_{SEB}$ at 15 ka. See Tab. 3 for peak ablation values.

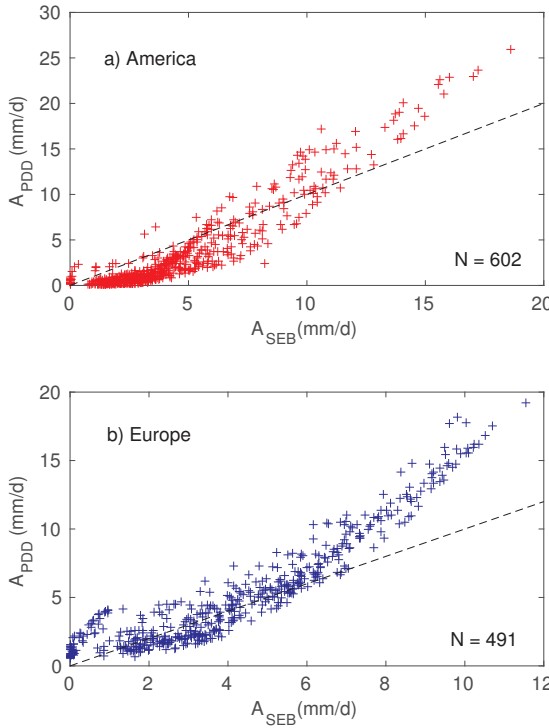

**Figure 8.** Scatter diagram of ablation rates by PDD method (offline) and SEB method from **(a)** American and **(b)** European ice sheets at 15 ka with equal NH ablation from both methods as shown in Fig. 7. $N$ is number of SICOPOLIS grid cells with non-zero ablation rate.




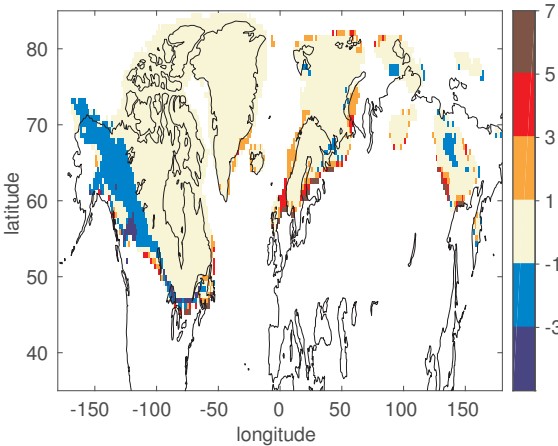

**Figure 9.** Geographic distribution of ablation differences (in $\mathrm{mm\,d^{-1}}$) obtained from PDD offline simulation using $\sigma=3\,^\circ\mathrm{C}$ and $(\alpha_S\,\alpha_I)=(9, 16)\,\mathrm{mm\,^\circ C^{-1}\,d^{-1}}$ relative to reference simulation at 15 ka, where NH total ablation from PDD and SEB methods agree closely (see Fig. 7). Thin black lines are present day topography.

**Table 1.** Mean anomaly ($m$) and rms–error ($r$) from 130 ka-long series for American and European ice sheets calculated from offline $A_{PDD}$ with temperature variability ($\sigma$) and melt factors ($\alpha_S, \alpha_I$) relative to $A_{SEB}$ as shown in Fig. 4. Note, $m$ is smallest for both ice sheets with standard PDD parameter values (**bold**) and $r$ is about factor three larger for American ice sheet than for European ice sheet.

| | | America | | Europe | |
|---|---|---|---|---|---|
| $\sigma$ | $(\alpha_S,\ \alpha_I)$ | $m$ | $r$ | $m$ | $r$ |
| $^\circ$C | $\mathrm{mm\,^\circ C^{-1}\,d^{-1}}$ | Sv | Sv | Sv | Sv |
| 3 | (5, 24) | -0.007 | 0.023 | 0.006 | 0.007 |
| 3 | (5, 20) | -0.011 | 0.023 | 0.002 | 0.005 |
| 3 | (5, 16) | -0.015 | 0.024 | -0.003 | 0.005 |
| 3 | (5, 12) | -0.019 | 0.025 | -0.007 | 0.007 |
| 3 | (5, 8) | -0.023 | 0.027 | -0.011 | 0.009 |
| 5 | (3, 12) | 0.008 | 0.021 | 0.008 | 0.010 |
| 5 | (3, 10) | 0.003 | 0.021 | 0.004 | 0.008 |
| **5** | **(3, 8)** | **-0.001** | 0.022 | **0.001** | 0.008 |
| 5 | (3, 6) | -0.006 | 0.024 | -0.003 | 0.008 |
| 5 | (3, 4) | -0.010 | 0.026 | -0.007 | 0.010 |





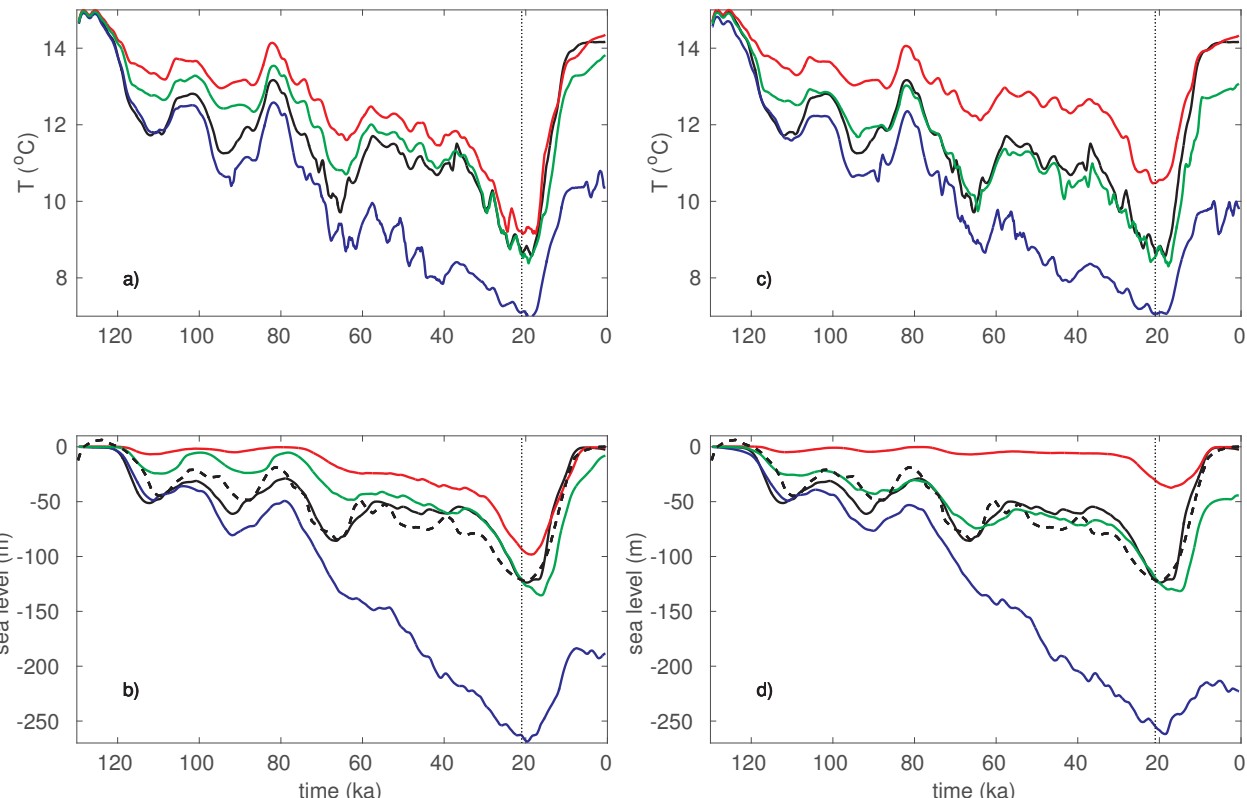

**Figure 10.** Glacial cycle simulations with online PDD method (**colored lines**) compared to reference simulation (**black continuous line**, cf. Fig. 1). **(a, c)** show global mean temperature and **(b, d)** show sea level together with reconstructed sea level (**black dashed line**). PDD-online simulations in **(a, b)** with $\sigma$=3 °C and in **(c, d)** with $\sigma$=5 °C reproduce climate closely either at inception (**blue lines**) or at termination (**red lines**) or at LGM (**green lines**). Melt factors $(\alpha_S, \alpha_I)$ in mm °C$^{-1}$ d$^{-1}$ used in **(a, b)** for simulations I3 (**blue**), T3 (**red**) and L3 (**green**) are (5,16), (9,16) and (7,20), respectively, and used in **(c, d)** for simulations I5 (**blue**), T5 (**red**) and L5 (**green**) are (3,8), (6,8) and (4,7), respectively. Note, simulation I5 uses standard PDD parameters and generates excessive cooling without recurrence to Holocene climate. Vertical dotted line marks 21 ka. See Tab. 4 for global mean $T$ and sea level at 21 and 0 ka.



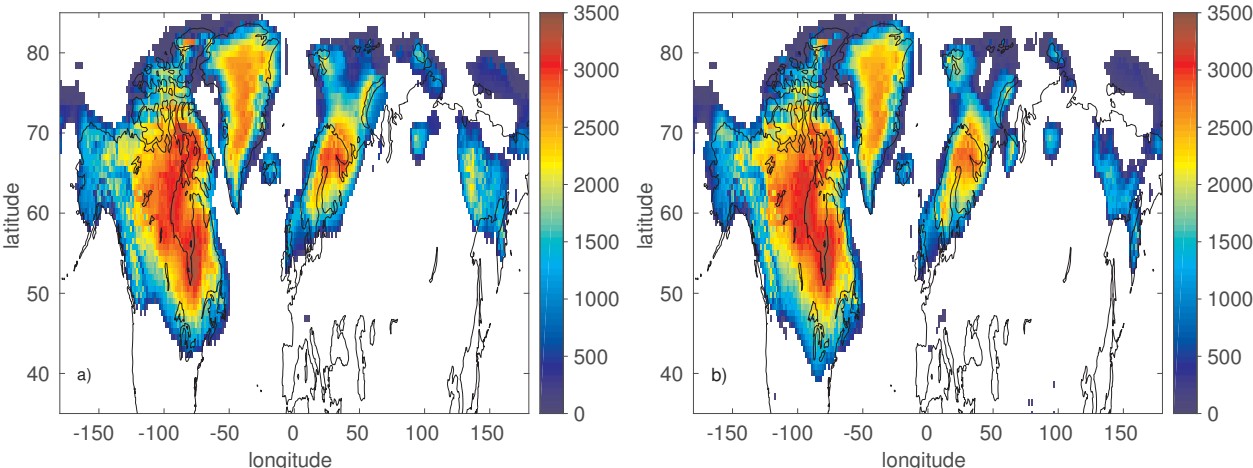

**Figure 11.** Simulated ice sheet thickness (in m) at 15 ka from **(a)** reference and **(b)** PDD-online simulation L3 which fulfills the LGM target window (see Tab. 4 and Fig. 10 for PDD parameter values). Thin black lines are present day topography.

**Table 2.** Summary of PDD parameters inducing minimum rms–errors between series of offline $A_{PDD}$ and $A_{SEB}$ for American and European ice sheets covering entire glacial cycle (see Fig. 5), glacial phase and glacial termination (see Fig. 6).

| | | America | Europe |
|---|---|---|---|
| interval | $\sigma$ | $(\alpha_S,\ \alpha_I)$ | $(\alpha_S,\ \alpha_I)$ |
| ka | °C | mm °C$^{-1}$d$^{-1}$ | mm °C$^{-1}$d$^{-1}$ |
| 130 – 0 | 3 | (10, 16) | (5, 16) |
| 130 –30 | 3 | ( 8, 16) | (5, 16) |
| 30 – 0 | 3 | (10, 16) | (6, 16) |
| 130 – 0 | 5 | (5, 12) | (3, 14) |
| 130 –30 | 5 | (4, 10) | (4, 6) |
| 30 – 0 | 5 | (6, 12) | (3, 16) |



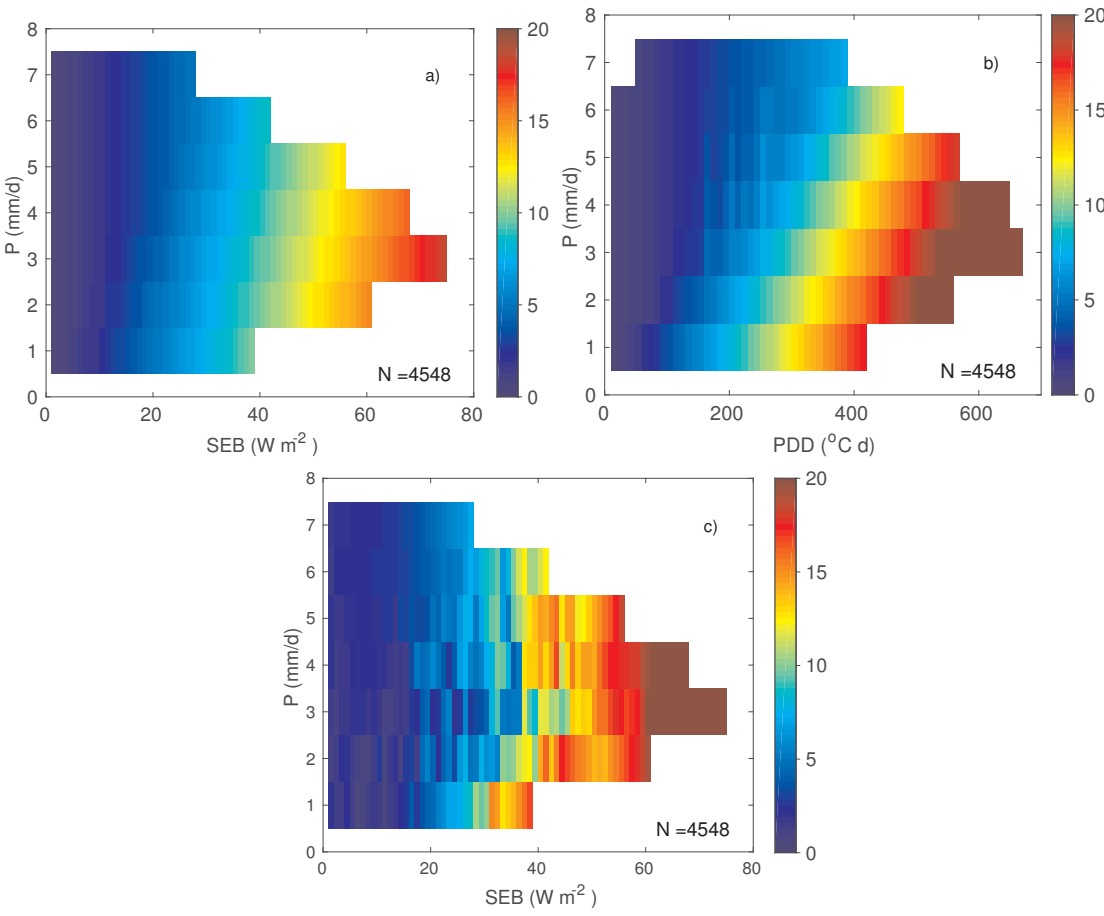

**Figure 12.** Comparison of characteristic dependencies of fine-resolution ablation rates (in $\mathrm{mm\,d^{-1}}$) showing in **(a)** $A_{SEB}$ as function of $SEB$ and $P$, in **(b)** $A_{PDD}$ (offline) as function of $PDD$ and $P$ and in **(c)** $A_{PDD}$ (offline) as function of $SEB$ and $P$. $A_{SEB}$ and $A_{PDD}$ are from NH ice sheets at 15 ka where NH total ablation from SEB approach and PDD offline method agree closely (see Fig. 7). Note that in **(c)** $A_{PDD}$ values between 5 and 15 $\mathrm{mm\,d^{-1}}$ vary irregular with corresponding $SEB$ values. $N$ is number of ice-covered grid cells.





**Table 3.** Ablation (in Sv) from NH total, American and European ice sheets at glacial termination (16–14 ka) where maximum in $A_{SEB}$ at 15 ka for NH is closely reproduced with offline PDD method using $\sigma$=3 $^\circ$C and $(\alpha_S, \alpha_I)$=(6, 19) in mm $^\circ$C$^{-1}$d$^{-1}$ (see Fig. 7). But maxima in ablation (**bold**) occur a millennium earlier in $A_{SEB}$ than in $A_{PDD}$ for NH total and American ice sheets. Note, while the total ablation at 15 ka from both method are close, $A_{SEB}$ in America is underestimated and $A_{SEB}$ in Europe is overestimated by the PDD method.

| time | NH | | America | | Europe | |
|---|---|---|---|---|---|---|
| ka | $A_{SEB}$ | $A_{PDD}$ | $A_{SEB}$ | $A_{PDD}$ | $A_{SEB}$ | $A_{PDD}$ |
| 16 | 0.37 | 0.32 | **0.25** | 0.18 | 0.07 | 0.08 |
| 15 | **0.41** | 0.42 | 0.24 | **0.19** | 0.12 | 0.16 |
| 14 | 0.38 | **0.45** | 0.16 | 0.14 | **0.18** | **0.24** |

**Table 4.** Global surface air temperature ($T$) and sea level ($sl$) at 21 ka (LGM) and 0 ka (MOD) from reference simulation (RS) compared with PDD-online simulations using $\sigma$ in $^\circ$C and $(\alpha_S, \alpha_I)$ in mm $^\circ$C$^{-1}$d$^{-1}$. PDD-online simulations are selected to fulfill the target windows glacial inception (I3, I5), glacial termination (T3, T5) and LGM (L3, L5) as shown in Fig. 10. Note, simulation I5 uses standard PDD parameter values (**bold**).

| name | $\sigma$ | $(\alpha_S, \alpha_I)$ | $T$ ($^\circ$C) | | $sl$ (m) | |
|---|---|---|---|---|---|---|
| | | | LGM | MOD | LGM | MOD |
| RS | | | 8.7 | 14.2 | -122 | -3.3 |
| I3 | 3 | ( 5, 16) | 7.1 | 10.6 | -263 | -189 |
| T3 | 3 | ( 9, 16) | 9.2 | 14.4 | -94 | -0.5 |
| L3 | 3 | ( 7, 20) | 8.7 | 13.9 | -121 | -7.5 |
| I5 | **5** | **( 3, 8)** | 7.1 | 9.2 | -255 | -223 |
| T5 | 5 | ( 6, 8) | 10.5 | 14.4 | -31 | -0.3 |
| L5 | 5 | ( 4, 7) | 8.5 | 13.1 | -120 | -45 |