# Peer review of "Comparison of surface mass balance of ice sheets simulated by positive-degree-day method and energy balance approach"

_Climate of the Past, 2016_

## Short Comment (SC1) · 14 Oct 2016

Really interesting paper. Did you do any sensitivity tests looking at the impact of ice sheet model resolution on the resulting mass balance (either PDD or SEB)? From what I can gather, the ice sheet surface mass balance is 1.5deg x 0.75deg. This seems somewhat coarse. van den Berg et al (2006, J. Glaciology) suggest that lower resolution surface mass balance can have a wide variety of solutions because slight changes in the ELA can have a large impact on the size of the ablation area. Other papers have shown similar results, in which the surface mass balance does not stabilize until resolutions finer than 10 km (Ullman et al., 2015; Nat. Geosci).

---

## Author Comment (AC1) · 28 Oct 2016

Thanks for pointing out the issue of resolution. Certainly resolution matters, in particular of the ice sheets, when modeling the ice sheet evolution. The present study is undertaken with a widely tested model version which is designed for long-term transient simulations of the climate evolution during the Quaternary. This Earth system model of intermediate complexity includes fully interactively coupled models for the atmosphere, the ocean, the vegetation and the ice sheets of the NH. Here transient simulations are used to contrast the SEB approach and PDD method rather than using equilibrium simulations. The simulations show that the two different descriptions of the ablation process lead to very different evolutions of the ice volume. We evaluated the

simulated ice volume changes against global sea level data. To our knowledge, data for evaluating the size of the ablation area over the time period of the glacial cycle are not available. Likely, having more observational data, the model can be developed further and processes can be described in more detail and with finer resolution. So, view this study simply as an exercise to test descriptions of the ablation process which meet basic requirements for the simulation of glacial-interglacial climate changes. This study is done with a type of forerunner model as a contribution for developing a more detailed and internally consistent Earth system model.

---

## Referee Comment (RC1) · Anonymous Referee #1 · 9 Nov 2016

**Review of Bauer and Ganopolski, 2016, CPD**

This study compares two different approaches to simulate the ablation component of the surface balance over ice sheets. As there is currently quite some discussion on different ablation or melt schemes, this is a timely and relevant research topic, and the results are suitable to be published in Climate of the Past. However, I do have several general, and many smaller comments that I think should be discussed before final publication in Climate of the Past.

**GENERAL COMMENTS**

1) The ablation simulated through the positive-degree-day (PDD) method is compared to the ablation as simulated by the surface energy balance approach (SEB) in the coupled climate and ice sheet model set-up. Another ablation scheme that is currently quite popular is the insolation-temperature-melt (ITM) approach (see for example, Robinson et al., 2011; Robinson and Goelzer, 2014). I understand that additionally assessing this melt scheme would be a lot of extra work, but this alternative approach should as least be mentioned and referred to in the discussion. One of the likely reasons why the PDD method cannot perfectly capture the SEB simulated ablation evolution over long time scales (such as a glacial cycle) could be because it does not account for insolation changes. The ITM method does include the effect of varying insolation on melt (although it might have other drawbacks). Please discuss.
2) I also miss a section in the introduction explaining the time period you focus on. Explain why the last glacial cycle, and give some background information. Introduce terms like inception, termination, LGM, and Holocene. And give dates for your "target windows", and call it "target periods" or similar.
3) Also lacking is a discussion of your reference simulation with respect to geological reconstructions of the ice sheets over the last glacial cycle, and other modelling approaches.
4) The set-up of the "offline" and "online" PDD ablation methods need to be explained in more detail (e.g. page 4, line 25)
5) Discussion resolutions: what is the effect of the rather large gridboxes used in CLIMBER and SICOPOLIS?
6) Also, the PDD method is originally developed for daily temperature input, as are the literature values for PDD factors and the standard deviation for temperature. You use 3-day mean temperatures. Please discuss.
7) What are the initial conditions for the (reference) simulation(s)? Same as pre-industrial? Does that mean only ice on Greenland (how much?), or where else? A map of the initial ice distribution for the reference simulation would be helpful.
8) Basing the selection of the PDD factors for the online simulations on the best PDD factors for the offline simulations is not very convincing. I though the whole point was to show that interactions/feedbacks between the climate and ice sheets are important. Why not test a range of factors,

and select the best through some statistical evaluation, such as the rms-error approach used for the offline simulations?

9) Some scientists do not have access to a climate model or not the computational resources to run it over long time scales, and therefore do not have access to SEB-derived ablation. Could you give a recommendation on how to best apply the PDD method. I.e. emphasize testing different PDD values, use a short time period, select one ice sheet, ...?

**SPECIFIC AND TECHNICAL COMMENTS**

Some sentences/sections are not very clear. I made several suggestions (see below), but would highly recommend a thorough English language check.

Abstract, lines 10-18: not clear, please rewrite. Make clear that you tested a range of literature values, and that it was not possible to find one set of PDD values that result in a good fit of both the American and the European ice sheets to your reference simulation. Neither can fixed values satisfactorily explain the ablation evolution over the entire glacial cycle for the individual ice sheets.

Abstract, line 18-19: change to: According to our simulations, the SEB approach is superior to the PDD methods when simulation Northern Hemisphere glacial cycles. This is partly due to the SEB approach including effects of change snow albedo, which is particularly important for the American ice sheet margins.

Page 1, line 20: change "gains and losses" to "fluctuations"

Page 2, line 1: Is it correct to say that the surface mass balance is the **main** factor affecting the evolution of ice sheets? What about calving, and basal processes? Needs a reference.

Page 2, line 6: rewrite "very close to each other".

Page 2, line 23: change to "demonstrated that feedbacks between climate ..."

Page 2, lines 24-31: Here add some more information on the ITM method

Page 2, line 32: change "problem" to "disadvantage"

Page 3, line 10: change to "Charbit et al. (2013) who discuss the effect of different PDD parameterizations on Northern Hemisphere ice evolution, we"

Page 3, line 15, change to "ice volume evolution"

Page 3, line 19-20, change to "evaluate the resulting glacial cycle simulations against sea level reconstructions."

Page 3, line 29: please explain what you mean with "balance year"

Page 5, line 18: delete "supposedly"

Page 6, line 12: change to "on North America and in Eurasia extending up to 120E. Note that the Greenland ice sheet is not included in the selections, but is part of the NH total.", or similar.

Page 6, lines 19-28: Please rewrite, could be shortened as well.

Page 7, line 3: The surface mass balance is also positive during periods of ice volume reductions (e.g. ~110ka, 90ka), indicating that a positive surface mass balance does not automatically lead to the build-up of ice. Please explain.

Page 7, line 6: change to "the Atlantic meridional overturning"

Page 7, line 11: change "control" to "tuneable"

Page 7, lines 28-30. Confusing to read, please rephrase.

Page 8, section 3.2: Why is the entire ensemble discussed for the rms-error, and only a selection for the anomaly/offset m? Figure 4 could be replaced by a figure similar to Figure 5, but than for the anomaly m. The information of the original Figure 4 can also be seen in Figure 6, especially if you add a (blue) line for the PDD-derived ablation evolution of the simulation that fit best to the reference simulation, over the entire 130ka.

Page 8, lines 18-19: Change to "Figure 6 shows the PDD-derived ablation evolution for the American and European ice sheets for the entire ensemble."

Page 8, lines 22-23: describe the "shorter" time intervals using "inception" and "deglaciation"

Page 8, lines 27-32: bit redundant, it was already clear from Figure 6 an Table 2 that different PDD factors are needed for different ice sheets. Maybe shorten?

Page 9, lines 1-9: make more clear that here the spatial patterns are investigated, not anymore the time evolution.

Page 9, lines 14-15: change to ".. the ice sheets are coupled through the PDD methods. In doing so, processes ignored by the PDD method, such as the impact from changing snow…"

Page 10: change "The simulation X" to "Simulation X"

Page 11, lines 8-16: Make clear that this discusses the offline simulations. Is it possible that the American ice sheet is less well simulated in the offline PDD method because the PDD scheme does not account for dust deposition?

Page 11, lines 17-19. Unclear, please rewrite. (blurred?)

Page 11, discussion of Figure 12 is also not clear, please rewrite.

Conclusions, lines 10-13: too technical. This means that different sets of PDD constants should be used depending on (1) the ice sheet, (2) the time period interested in. Right?

Figure 1: Some indication of the orbital forcing should be included. Maybe summer 65N insolation or the Milankovitch parameters precession/tilt?

Figure 9: change "topography" to "coastlines"

Figure 11: change "15ka" to "21ka" and "topography" to "coastlines"

**REFERENCES**

Robinson, A., Calov, R., and Ganopolski, A.: Greenland ice sheet model parameters constrained using simulations of the Eemian Interglacial, Clim. Past., 7, 381-396, doi:10.5194/cp-7-381-2011, 2011.

Robinson, A. and Goelzer, H.: The importance of insolation changes for paleo ice sheet modeling, The Cryosphere, 8, 1419-1428, doi:10.5194/tc-8-1419-2014, 2014.

---

## Referee Comment (RC2) · Anonymous Referee #2 · 30 Nov 2016

The study aims at comparing the northern hemispheric surface mass balance throughout the last glacial cycle on basis of a positive-degree-day (PDD) and a surface-energy-balance (SEB) approach. While the SEB approach allows for a realistic representation of the last glacial cycle, as compared to reconstructions (e.g. sea level), the PDD approach shows significant shortcomings if constant values for melt factors and short-term temperature variability are used. The authors discuss a very relevant and timely research topic, which is important for the understanding of the role of ice sheets within the climate system. While the paper is well structured and the simulations are interesting and insightful, revisions are required to improve the comprehensibility.

[Figure]

**Page 1, Line 10 to 18**: This paragraph of the abstract is somewhat confusing. It would be good if the authors could revise this section; I would suggest either by explaining the simulation setups in more detail or by putting more emphasize on the results and less on the simulations setup, given that they will introduce the setups in detail later.

**Page 3, Line 17 to 20**: Please introduce here the "offline" and "online" PDD approaches. This will help to understand what is meant by those two approaches (as they are not explicitly mentioned in the Section "Model description"). To understand the difference is crucial for interpreting the results.

**Page 4, Section 2.2**: The PDD approach is described in detail but the SEB approach is only briefly mentioned. Although the reference Calov et al. (2005) is given additional information regarding the setup would be useful. How is the downscaling from the 7x18 atmospheric grid to the higher resolution SICOPOLIS grid done? How are certain processes regarded when downscaling (e.g. height desertification effect)? Further, it would be good to mention that a one-layer snow model is used. Please also introduce the parameterization of the albedo, given that changes in the albedo of the ice sheet seem to be crucial for the simulation of the last glacial cycle.

**Page 6, Line 29-31**: While discussing the differences between the American and European ice sheet I am wondering how well CLIMBER represents the interactions between the two ice sheets. Previous studies (e.g. Liakka et al., 2016) have shown that the European ice sheet is significantly influenced by the American Ice Sheet. While discussing reasons for the different responses of the European and American ice sheets these processes should be shortly discussed in regards to the presented results.

**Page 12, Discussion**: While the results clearly indicate that the SEB approach is superior to the PDD approach for simulating the last glacial cycle it would be good to point towards the weaknesses of both approaches. This might be covered by a more detailed description of the SEB in the method section (see above) or one or two sentences in the discussion section. Further, how realistic are the SEB results?

Most of the results are integrated over the Northern Hemisphere but how is the spatial distribution? It could be good to see e.g. a comparison between the ice sheets derived with the SEB approach during LGM in comparison to LGM reconstructions on a spatial map.

**Page 11, Line 14-16 and Conclusions**: The authors state that the American melt depends largely on the snow melt factor, which can be attributed to the effect of dust deposition. I think the authors need to clarify how dust deposition and snow age interplay in the model. Is the albedo change a linear function of the snow age/dust or do other factors play in? What is the relationship between snow age (simply changes of snow properties) and dust deposition? Could it be other factors that cause these differences?

**Page 11, Line 25 to 31**: Fig. 12 needs to be explained better. Please clarify this paragraph. Currently it is hard to follow the reasoning.

**Minor issues**:

**Page 1, Line 2**: precessional

**Page 2, Line 20**: ... meteorological conditions on high frequency time scales – the difference of the input data between SEB and PDD is not clear.

**Page 7 and 8**: There is a mismatch between the figure order as mentioned in the text and the actual figure order. Fig. 6 before Fig 4 and 5.

**Page 7, Line 18 to 19**: Remove second 'lie in the range'. Repetition.

**Page 7, Line 23**: Remove "range". Repetition.

**Page 7, Line 30**: "." Before "Thereby".

**Page 9, Line 25**: "we use the latter alphaI value and vary alphaS"

[Figure]

**Several times throughout the text**: "vice versa" and not "vice verse"

**Fig. 9 and 11**: The authors could consider a more realistic map projection.

**Throughout the text**: Please revise for language mistakes.

---

## Author Comment (AC2) · 16 Dec 2016

**Reviewer 1**

*GENERAL COMMENTS*

*1. The ablation simulated through the positive-degree-day (PDD) method is compared to the ablation as simulated by the surface energy balance approach (SEB) in the coupled climate and ice sheet model set-up. Another ablation scheme that is currently quite popular is the insolation-- temperature--melt (ITM) approach (see for example, Robinson et al., 2011; Robinson and Goelzer, 2014). I understand that additionally assessing this melt scheme would be a lot of extra work, but this alternative approach should as least be mentioned and referred to in the discussion. One of the likely reasons why the PDD method cannot perfectly capture the SEB simulated ablation evolution over long time scales (such as a glacial cycle) could be because it does not account for insolation changes. The ITM method does include the effect of varying insolation on melt (although it might have other drawbacks). Please discuss.*

Indeed, several years ago we developed the regional model REMBO which is based on ITM approach and we used REMBO in a number of publications. However, it is important to note that REMBO was specially designed for Greenland and for climate conditions which are not very different from present. The ITM scheme contains apart from two empirical parameters, which are likely spatially and temporally dependent, one parameter – transparency of the atmosphere - which is known to vary strongly spatially and in time. We have no idea how ITM can be parameterized for the purpose of simulations of large scale glaciations during entire glacial cycles. Therefore we never used ITM for this purpose. And, although, ITM does have some advantages over the PDD approach, we do not believe that ITM can be considered as the real alternative to the physically based SEB approach.

*2) I also miss a section in the introduction explaining the time period you focus on. Explain why the last glacial cycle, and give some background information. Introduce terms like inception, termination, LGM, and Holocene. And give dates for your "target windows", and call it "target periods" or similar.*

*3) Also lacking is a discussion of your reference simulation with respect to geological reconstructions of the ice sheets over the last glacial cycle, and other modelling approaches.*

The choice of the last glacial cycle is rather obvious – it is best covered by paleoclimate records, especially since the LGM. This is why most of previous modeling study of glacial cycles have been performed for the last glacial cycle. As far as our model performance for the glacial cycle is concerned (reference run), it has been described in detail and compared with available climatological data in Ganopolski et al (2010). Our reference run is practically identical to that model which is analyzed in Ganopolski et al (2010). We just refer to that paper which was published in open access journal and readily available for any reader.

*4) The set-up of the "offline" and "online" PDD ablation methods need to be explained in more detail (e.g. page 4, line 25).*

We will describe the difference between  "offline" and "online" simulations in section 2.3 more clearly.

5)  *Discussion resolutions: what is the effect of the rather large grid boxes used in CLIMBER and SICOPOLIS?*

We do not believe that the manuscript under consideration is the right one for discussing the resolution issue. We and other groups around the world have already published ca. 200 papers based on  CLIMBER-2 model. In many of those papers an extensive comparison of CLIMBER-2 results with observed present, reconstructed past and simulated future climates by GCMs is presented. These studies revealed that on its resolution CLIMBER-2 is doing a reasonably good job. The coupling between the coarse resolution climate component of CLIMBER-2 and the relatively high resolution (70km) ice sheet component is, indeed, a nontrivial task to which we devoted significant efforts. The coupling is based on spatial and vertical interpolation and, additionally, parameterization of sub-grid processes, such as orographic precipitation. This is described in detail in Calov et al (2005) and Ganopolski et al (2010). Obviously, using a higher resolution is always desirable but for simulations of glacial cycles a high spatial resolution is costly. At present the CLIMBER-2 model is the only comprehensive Earth system model which is able to simulate numerous glacial cycles. Therefore we cannot compare it with the results of higher resolution models.   For readers not familiar with previous works made with CLIMBER-2 we will add a paragraph in the Introduction section discussing potential caveats related to coarse spatial resolution of our model.

*6)  Also, the PDD method is originally developed for daily temperature input, as are the literature values for PDD factors and the standard deviation for temperature. You use 3 - day mean temperatures. Please discuss.*

In fact, PDD methods are developed for using climatological monthly temperatures which are then interpolated to produce daily temperatures. Therefore calculation of PDD by use of a 1-day or 3-day time step produces essentially the same result.  In CLIMBER-2, the time step in the physically based EBM is three days. This is done to reduce computational cost.  This is why we used the same 3-day time step for calculation of PDD. Of course, the factor 3 was taken into account when we calculated PDD. This issue will be clarified in the revised manuscript.

*7)  What are the initial conditions for the (reference) simulation(s)? Same as pre-industrial? Does that mean only ice on Greenland (how much?), or  where else? A map of the inital ice distribution for the reference simulation would be helpful.*

In all our experiments the equilibrium state of the climate-cryosphere system obtained for present-day conditions was used as initial condition and the model was run from 130 kyr BP until the present. We now clarify this in the text. Since the simulated preindustrial climate looks very much alike the observed present-day state containing the Greenland ice sheet as the only ice sheet in NH, we do not believe that such a figure would be very useful.

8)  *Basing the selection of the PDD factors for the online simulations on the best PDD factors for the offline simulations is not very convincing. I though the whole point was  to show that interactions/feedbacks between the climate and ice sheets are important. Why not test a range of factors, and select the best through some statistical evaluation, such as the rms-- error approach used for the offline simulations?*

First, we believe that using of the "best" PDD factors found in offline simulations for online simulations is a rather natural choice. Second, importance of feedbacks between climate and ice sheet is well known and is not the point of our study. The main result of our study is that it is not possible to find a pair of PDD factors which are suitable for simulation of the entire glacial cycle. We would like to clearly say that we did not try to find the best PDD factors which we would recommend to other modelers to use. To the contrary, the conclusion of our

paper is very clear – we do not recommend to use the PDD approach for simulating glacial cycles. And this is related to the last general comment:

*Some scientists do not have access to a climate model or not the computational resources to run it over long time scales, and therefore do not have access to SEB - derived ablation. Could you give a recommendation on how to best apply the PDD method. I.e. emphasize testing different PDD values, use a short time period, select one ice sheet, …?*

Our study shows that a realistic simulation of an entire glacial cycle with the same PDD parameters is not possible. Sure, one can pursue a kind of inverse modeling approach to infer PDD parameters for different time intervals to obtain results comparable with paleoclimate reconstructions. However, the scientific value of such modeling is questionable. As the modeling of individual aspects of a glacial cycle, such as glacial inception or termination concerns, a "recommendation" according to our study is clear. Namely, to simulate glacial inception one has to use smaller PDD values than for simulating glacial termination. But, again, since we believe that our study convincingly demonstrates that PDD is not an adequate method for modeling the ice sheet surface mass balance during glacial cycles, we are reluctant to give any explicit recommendation of how to "improve" the PDD approach.

*SPECIFIC AND TECHNICAL COMMENTS*

We will address all comments appropriately.

The Figure 4 will be replaced and Figure 6 will be modified according to the suggestions.

*Page 8, section 3.2: Why is the entire ensemble discussed for the rms-error, and only a selection for the anomaly/offset m? Figure 4 could be replaced by a figure similar to Figure 5, but than for the anomaly m. The information of the original Figure 4 can also be seen in Figure 6, especially if you add a (blue) line for the PDD-derived ablation evolution of the simulation that fit best to the reference simulation, over the entire 130ka.*

Figure 4 will show the bivariate distributions of the mean anomaly and the rms-error calculated for the total NH ice sheet. The new Figure will show that the minimum in anomaly is not constrained by a unique pair of melt factors.

*Page 11, discussion of Figure 12 is also not clear , please rewrite.*

Figure 12 will be replaced by a new Figure showing time series of insolation and ablation for June and July, as in June insolation is largest and in July ablation is largest. We will present the simulated ablation with the SEB approach in response to snow albedo changes induced separately by snow aging and dust deposition. The new Figures will show results from offline and online simulations and which will be described in the Discussion section.

---

## Author Comment (AC3) · 16 Dec 2016

**Reviewer 2**

*Page 1, Line 10 to 18: This paragraph of the abstract is somewhat confusing. It would be good if the authors could revise this section; I would suggest either by explaining the simulation setups in more detail or by putting more emphasize on the results and less on the simulations setup, given that they will introduce the setups in detail later.*

Abstract will be revised.

*Page 3, Line 17 to 20: Please introduce here the "offline" and "online" PDD approaches. This will help to understand what is meant by those two approaches (as they are not explicitly mentioned in the Section "Model description"). To understand the difference is crucial for interpreting the results.*

We will describe the difference between "offline" and "online" simulations in section 2.3 "Positive-degree-day (PDD) method" and in the Discussion section.

*Page 4, Section 2.2: The PDD approach is described in detail but the SEB approach is only briefly mentioned. Although the reference Calov et al. (2005) is given additional information regarding the setup would be useful. How is the downscaling from the 7x18 atmospheric grid to the higher resolution SICOPOLIS grid done? How are certain processes regarded when downscaling (e.g. height desertification effect)? Further, it would be good to mention that a one-layer snow model is used. Please also introduce the parameterization of the albedo, given that changes in the albedo of the ice sheet seem to be crucial for the simulation of the last glacial cycle.*

A detailed description of the surface energy and mass balance scheme (SEMI) is given in Calov et al. (2015) and it is not possible to repeat all details here. However, for readers' convenience we will add a paragraph where we briefly describe the coupling procedure and major parameterizations.

*Page 6, Line 29-31: While discussing the differences between the American and European ice sheet I am wondering how well CLIMBER represents the interactions between the two ice sheets. Previous studies (e.g. Liakka et al., 2016) have shown that the European ice sheet is significantly influenced by the American Ice Sheet. While dis- cussing reasons for the different responses of the*

*European and American ice sheets these processes should be shortly discussed in regards to the presented results.*

It is difficult to compare our modeling results with Liakka et al. (2016) because they performed equilibrium time slice experiments while we performed transient experiments. In the model running over the orbital time scales, ice sheets are never in equilibrium with climate. In our simulations, the Laurentide ice sheet does exert a strong cooling over the North Atlantic and significantly influences the European climate. However, it is important to note that due to coarse spatial resolution of CLIMBER-2, we only account for thermally driven atmospheric stationary waves but not for topographically forced. The omission of the latter may affect long-distance climate teleconnections.

*Page 12, Discussion: While the results clearly indicate that the SEB approach is superior to the PDD approach for simulating the last glacial cycle it would be good to point towards the weaknesses of both approaches. This might be covered by a more detailed description of the SEB in the method section (see above) or one or two sentences in the discussion section. Further, how realistic are the SEB results? Most of the results are integrated over the Northern Hemisphere but how is the spatial distribution? It could be good to see e.g. a comparison between the ice sheets derived with the SEB approach during LGM in comparison to LGM reconstructions on a spatial map.*

We believe that weaknesses of the PDD approach are obvious from our study. The SEB approach is entirely physically based and therefore the only right but of course, its implementation in the model, which does not simulate synoptic and intra-annual climate variability, requires a number of assumptions and additional parameterizations. We will discuss this is the Discussion section. As far as the performance of our standard run is concerned (including spatial distribution of ice sheets) it is discussed in detail in Ganopolski et al. (2010).

*Page 11, Line 14-16 and Conclusions: The authors state that the American melt depends largely on the snow melt factor, which can be attributed to the effect of dust deposition. I think the authors need to clarify how dust deposition and snow age interplay in the model. Is the albedo change a linear function of the snow age/dust or do other factors play in? What is the relationship between*

*snow age (simply changes of snow properties) and dust deposition? Could it be other factors that cause these differences?*

The albedo scheme is described in Calov et al (2005). Indeed, there is an interplay between aging and the effect of dust on snow albedo – the latter is stronger for the "old" show. We add further information in the Discussion section. We will insert a new Figure 12 (thereby replacing the former Figure 12) to show ablation series in response to the aging effect of pure snow and the aging effect of impure snow.

**Page 11, Line 25 to 31**: *Fig. 12 needs to be explained better. Please clarify this paragraph. Currently it is hard to follow the reasoning.*

The former Figure 12 will be removed. Please see our response to the comment above and also to Reviewer#1.

*Minor Issues*

The manuscript will be revised according to all minor issues.

---

## Author Response (AR1)

We would like to thank both reviewers for their very helpful comments and suggestions. We significantly revised the manuscript following reviewers' comments and suggestions. In particular we

- replaced  Figure 4 by the new one
- discussed ITM method,  its strengths and weaknesses
- explained  difference between offline and online simulations
- add entire new section (5) describing new experiments which shed light on importance of snow albedo parameterization for modeling of ice sheets evolution during glacial cycles

In the following, we provide specific responses to each of the points raised in reviewers comments.

**Reviewer 1**

*GENERAL COMMENTS*

*1. The ablation simulated through the positive-degree-day (PDD) method is compared to the ablation as simulated by the surface energy balance approach (SEB) in the coupled climate and ice sheet model set-up. Another ablation scheme that is currently quite popular is the insolation-- temperature--melt  (ITM) approach (see for example, Robinson et al., 2011; Robinson and Goelzer, 2014). I understand  that additionally assessing   this melt scheme would be a lot of extra work, but this alternative  approach should at least be mentioned and referred to in the discussion. One of the likely reasons why the PDD method cannot perfectly capture the SEB simulated ablation evolution over long time scales (such as a glacial cycle) could be because it does not account for insolation changes. The ITM method does include the effect of varying insolation on melt (although it might have other drawbacks). Please discuss.*

Indeed, several years ago we developed the regional model REMBO which is based on ITM approach and we used REMBO in a number of our publications. However, it is important to note that REMBO was specially designed for Greenland and for climate conditions which are not very different from present. The ITM scheme contains apart from two empirical parameters, which are likely spatially and temporally dependent, one parameter – transparency of the atmosphere - which is known to vary strongly spatially and in time. We have no idea how ITM can be parameterized for the purpose of simulations of large scale glaciations during entire glacial cycles. Therefore we never used ITM for this purpose. And, although, ITM does have some advantages over the PDD approach, we do not believe that ITM can be considered as the real alternative to the physically based SEB approach.

*2) I also miss a section in the introduction explaining the time period you focus on. Explain why the last glacial cycle, and give some background information. Introduce terms like inception, termination, LGM, and Holocene. And give dates for your "target windows", and call it "target periods" or similar.*

*3) Also lacking is a discussion of your reference simulation with respect to geological reconstructions of the ice sheets over the last glacial cycle, and other modelling approaches.*

The choice of the last glacial cycle is rather obvious – it is period of time best covered by paleoclimate records, especially since the LGM. This is why most of previous modeling study of glacial cycles has been performed for the last glacial cycle. As far as our model performance for the glacial cycle is concerned (reference run), it has been described in detail and compared with available climatological data in Ganopolski et al. (2010). Our reference run is practically identical to that model which is analyzed in Ganopolski et al. (2010). We just refer to that paper which was published in open access journal and is readily available for any reader.

*4) The set-up of the "offline" and "online" PDD ablation methods need to be explained in more detail (e.g. page 4, line 25).*

We agree that its was not clear described.  Now we devoted entire paragraph (last paragraph, section 2.3) describing the difference between  "offline" and "online" simulations.

5)  *Discussion resolutions: what is the effect of the rather large grid boxes used in CLIMBER and SICOPOLIS?*

We do not believe that the manuscript under consideration is the right one for discussing the resolution issue. In a number of our papers we presented  an extensive comparison of CLIMBER-2 results with observed present, reconstructed past and simulated future climates by GCMs. These studies revealed that on its very coarse grid CLIMBER-2 does a reasonably good job. The coupling between the coarse resolution climate component of CLIMBER-2 and the relatively high resolution (70km) ice sheet component  is, indeed, a nontrivial task to which we devoted significant efforts. The coupling is based on spatial and vertical interpolation and, additionally, parameterization of sub-grid processes, such as orographic precipitation. How it is done is described in detail in Calov et al. (2005) and Ganopolski et al. (2010). Obviously, using a higher spatial resolution is always desirable but for simulations of glacial cycles a high spatial resolution is costly. At present, the CLIMBER-2 model is the only comprehensive Earth system model which is able to simulate numerous glacial cycles. Therefore we cannot compare performance of our model with higher resolution climate-ice sheets models on the orbital time scales .  However, coarse spatial resolution of atmospheric component of CLMBER-2 is obvious limitation of our study  and we admit  this fact in the conclusions.

*6)  Also, the PDD method is originally developed for daily temperature input, as are the literature values for PDD factors and the standard deviation for temperature. You use 3 - day mean temperatures. Please discuss.*

In fact, PDD methods are developed for using climatological monthly temperatures which are then interpolated to produce daily temperatures. Therefore calculation of PDD by using of a 1-day or 3-day time step produces essentially the same result. Since in CLIMBER-2, the time step in the physically-based SEB surface mass balance scheme is three days, we used the same time step for calculation of PDD. Of course, the factor 3 was taken into account when we computed sum of positive degree days.

*7) What are the initial conditions for the (reference) simulation(s)? Same as pre-industrial? Does that mean only ice on Greenland (how much?), or where else? A map of the inital ice distribution for the reference simulation would be helpful.*

In all our experiments, equilibrium state of the climate-cryosphere system obtained for present-day conditions was used as initial condition and the model was run from 130 ka until present. We now clarify this in the text. Since initial distribution of ice sheets is very much alike the observed present-day state with the Greenland ice sheet being the only ice sheet in NH, we do not believe that such a figure would be very useful.

8) *Basing the selection of the PDD factors for the online simulations on the best PDD factors for the offline simulations is not very convincing. I though the whole point was to show that interactions/feedbacks between the climate and ice sheets are important. Why not test a range of factors, and select the best through some statistical evaluation, such as the rms-- error approach used for the offline simulations?*

First, computational cost of offline simulations is small compare to the online simulations and we cannot perform online simulations with all possible combinations of melt parameters as we've done in offline simulations. Therefore we believe that using of the "best" PDD factors found in offline simulations is a rather natural choice for online simulations. Second, we believe that out study (in particular Figure 10) nicely illustrates importance of feedbacks between climate and ice sheet. For example, for $\sigma$=5$^o$C, change of snow melt parameter from 3 to 6 mm C$^{-1}$ d$^{-1}$ leads to almost ten-fold decrease in LGM ice volume. However, the main result of our study is that it is not possible to find a set of three PDD factors which are suitable for simulation of the entire glacial cycle. We did not try to find the best PDD factors which we would recommend to other modelers to use. To the contrary, the conclusion of our paper is very clear – we do not recommend to use PDD approach for simulations glacial cycles. And this is related to the last general comment:

*Some scientists do not have access to a climate model or not the computational resources to run it over long time scales, and therefore do not have access to SEB - derived ablation. Could you give a recommendation on how to best apply the PDD method. I.e. emphasize testing different PDD values, use a short time period, select one ice sheet, …?*

Our study shows that a realistic simulation of the entire glacial cycle with the same PDD parameters is not possible. Sure, one can pursue a kind of inverse modeling approach to infer PDD parameters for different time intervals and ice sheets (even different latitudes or elevations) to obtain results comparable with paleoclimate reconstructions. However, the scientific value of such modeling is questionable. In the case of modeling of individual aspects of a glacial cycle, such as glacial inception or glacial termination, the conclusions of our study is quite clear. Namely, to simulate glacial inception one has to use smaller PDD parameters values than for simulation of glacial termination.

**SPECIFIC AND TECHNICAL COMMENTS**

*Abstract, lines 10---18: not clear, please rewrite. Make clear that you tested a range of literature values, and that it was not possible to find one set of PDD values that result in a good fit of both the American and the European ice sheets to your reference simulation. Neither can fixed values satisfactorily explain the ablation evolution over the entire glacial cycle for the individual ice sheets.*

The abstract has been almost completely rewritten. We now stated clearly what reviewer suggested. We only omitted mentioning of "standard literature values" because these values exist mostly only for Greenland and there is no reason why they should be suitable for very different ice sheets during glacial cycle. Therefore in offline simulations we explore a much broader range, but still came to the same conclusion – there is no single set of PDD parameters which is suitable for both major ice sheets during the entire glacial cycle.

*Abstract, line 18---19: change to: According to our simulations, the SEB approach is superior to the PDD methods when simulation Northern Hemisphere glacial cycles. This is partly due to the SEB approach including effects of change snow albedo, which is particularly important for the American ice sheet margins.*

We now reformulate the last sentence of the abstract as following: "According to our simulations, the SEB approach, including effects of changing snow albedo from dust deposition and aging, proves superior for simulation of glacial cycles"

*Page 1, line 20: change "gains and losses" to "fluctuations"*

Done

*Page 2, line 1: Is it correct to say that the surface mass balance is the main factor affecting the evolution of ice sheets? What about calving, and basal processes? Needs a reference.*

This sentence was removed.

*Page 2, line 6: rewrite "very close to each other".*

This sentence was modified.

*Page 2, line 23: change to "demonstrated that feedbacks between climate …"*

Done

*Page 2, lines 24---31: Here add some more information on the ITM method*

Now we added (p. 3) discussion of the ITM scheme

*Page 2, line 32: change "problem" to "disadvantage"*

Done

*Page 3, line 10: change to "Charbit et al. (2013) who discuss the effect of different PDD parameterizations on Northern Hemisphere ice evolution, we"*

Done

*Page 3, line 29: please explain what you mean with "balance year"*

The term "balance years" has been removed from the manuscript

*Page 5, line 18: delete "supposedly"*

Done

*Page 6, line 12: change to "on North America and in Eurasia extending up to 120E. Note that the Greenland ice sheet is not included in the selections, but is part of the NH total.", or similar.*

Done

*Page 6, lines 19---28: Please rewrite, could be shortened as well.*

Done.  This paragraph has been shortened significantly.

*Page 7, line 3: The surface mass balance is also positive during periods of ice volume reductions (e.g. ~110ka, 90ka), indicating that a positive surface mass balance does not automatically lead to the build-up of ice. Please explain*.

Now we reformulated this sentence as following:  "The resulting surface mass balance  (Fig. 2f) is positive and exceeds calving rate (not shown) during most of the glacial cycle leading   to the buildup of large ice sheets at the LGM." We agree that positive mass balance is not sufficient condition for ice sheet growth. For that mass balance should exceed calving rate. During several periods when ice volume is decreasing (prior to glacial termination), mass balance is positive but smaller than calving.

*Page 7, line 6: change to "the Atlantic meridional overturning"*

This sentence was removed but on the next page we now use the correct term - "Atlantic meridional overturning circulation"

Page 7, line 11: change "control" to "tunable"

Done

*Page 7, lines 28---30. Confusing to read, please rephrase.*

This sentence was reformulated. We hope that it is more clear now.

*Page 8, section 3.2: Why is the entire ensemble discussed for the rms---error, and only a selection for the anomaly/offset m? Figure 4 could be replaced by a figure similar to Figure 5, but than for the anomaly m. The information of the original Figure 4 can also be seen in Figure 6, especially if you add a (blue) line for the PDD---derived ablation evolution of the simulation that fit best to the reference simulation, over the entire 130ka.*

New Fig. 4 (which replaces the old Fig. 4) shows the bivariate distributions of the mean anomaly and the rms-error calculated for the total NH ice sheet. It shows that the minimum of absolute value of anomaly m (m=0) does not determine to a unique pair of melt factors. At the same time, as seen from comparison of m-plots and rms-plots, for both sigma values minimum in rms is located close to zero value of m. This justifies our choice of minimum in rms as the criteria for selection of the "optimal" values of melt factors.

*Page 8, lines 18---19: Change to "Figure 6 shows the PDD---derived ablation evolution for the American and European ice sheets for the entire ensemble."*

Done

*Page 8, lines 22---23: describe the "shorter" time intervals using "inception" and "deglaciation"*

The word "shorter" was removed. Indeed, interval 130-30 ka is not "short" but it also cannot be named "glacial inception". Similarly, 30-0 ka is not glacial termination.

*Page 8, lines 27---32: bit redundant, it was already clear from Figure 6 an Table 2 that different PDD factors are needed for different ice sheets. Maybe shorten?*

Done. The paragraph has been shortened considerably.

*Page 9, lines 1---9: make more clear that here the spatial patterns are investigated, not anymore the time evolution.*

Done. We added inserted "spatial patterns" in this sentence

*Page 9, lines 14---15: change to ".. the ice sheets are coupled through the PDD methods. In doing so, processes ignored by the PDD method, such as the impact from changing snow…"*

The sentence "In this way … is ignored" was removed

*Page 10: change "The simulation X" to "Simulation X"*

Done

*Page 11, lines 8---16: Make clear that this discusses the offline simulations. Is it possible that the American ice sheet is less well simulated in the offline PDD method because the PDD scheme does not account for dust deposition?*

Discussion was completely rewritten. The role of eolian dust is discussed in the new section 5.

*Page 11, lines 17---19. Unclear, please rewrite. (blurred?)*

This sentence was removed

*Page 11, discussion of Figure 12 is also not clear, please rewrite.*

Old Figure 12 was replaced by a new Figure 12 showing time series of insolation and ablation for June and July, as in June insolation is largest and in July ablation is largest. New Figure 13 shows results from online simulations which illustrate high sensitivity of simulated ice sheet to parameterization of snow albedo.

*Conclusions, lines 10---13: too technical. This means that different sets of PDD constants should be used depending on (1) the ice sheet, (2) the time period interested in. Right?*

This section was completely rewritten. Actually, we do not recommend to use different PDD parameter values for different ice sheets and for different periods of time, because scientific values of simulations with explicitly time-dependent model parameters is rather questionable. Instead we recommend to use solely SEB approach.

*Figure 9: change "topography" to "coastlines"*

Done

*Figure 11: change "15ka" to "21ka" and "topography" to "coastlines"*

Done

**Reviewer 2**

*Page 1, Line 10 to 18: This paragraph of the abstract is somewhat confusing. It would be good if the authors could revise this section; I would suggest either by explaining the simulation setups in more detail or by putting more emphasize on the results and less on the simulations setup, given that they will introduce the setups in detail later.*

Abstract was completely rewritten.

*Page 3, Line 17 to 20: Please introduce here the "offline" and "online" PDD a pproaches. This will help to understand what is meant by those two approaches (as they are not explicitly mentioned in the Section "Model description"). To understand the difference is crucial for interpreting the results.*

We now describe the difference between "offline" and "online" simulations in section 2.3 .

*Page 4, Section 2.2: The PDD approach is described in detail but the SEB approach is only briefly mentioned. Although the reference Calov et al. (2005) is given additional information regarding the setup would be useful. How is the downscaling from the 7x18 atmospheric grid to the higher resolution SICOPOLIS grid done? How are certain processes regarded when downscaling (e.g. height desertification effect)? Further, it would be good to mention that a one-layer snow model is used. Please also introduce the parameterization of the albedo, given that changes in the albedo of the ice sheet seem to be crucial for the simulation of the last glacial cycle.*

A detailed description of the surface energy and mass balance scheme (SEMI) is given in Calov et al. (2015) and it is not possible to repeat all details here. However, for readers' convenience we added a paragraph where we briefly describe the coupling procedure.

*Page 6, Line 29-31: While discussing the differences between the American and European ice sheet I am wondering how well CLIMBER represents the interactions between the two ice sheets. Previous studies (e.g. Liakka et al., 2016) have shown that the European ice sheet is significantly influenced by the American Ice Sheet. While discussing reasons for the different responses of the European and American ice sheets these processes should be shortly discussed in regards to the presented results.*

It is difficult to compare our modeling results with Liakka et al. (2016) because they performed equilibrium time slice experiments while we performed transient experiments. In the model running over the orbital time scales, ice sheets are never in equilibrium with climate. In our simulations, the Laurentide ice sheet does exert a strong cooling over the North Atlantic and significantly influences the European climate. However, it is important to note that due to coarse spatial resolution of CLIMBER-2, we only account for thermally driven atmospheric stationary waves but not for topographically forced waves. The omission of the latter affects long-distance climate teleconnections.

*Page 12, Discussion: While the results clearly indicate that the SEB approach is superior to the PDD approach for simulating the last glacial cycle it would be good to point towards the weaknesses of both approaches. This might be covered by a more detailed description of the SEB in the method section (see above) or one or two sentences in the discussion section. Further, how realistic are the SEB results? Most of the results are integrated over the Northern Hemisphere but how*

*is the spatial distribution? It could be good to see e.g. a comparison between the ice sheets derived with the SEB approach during LGM in comparison to LGM reconstructions on a spatial map.*

We believe that weaknesses of the PDD approach are obvious from our study. The SEB approach is entirely physically based and therefore the only right but of course, its implementation in the model, which does not simulate synoptic and intra-annual climate variability, requires a number of assumptions and additional parameterizations. We discuss this is the Discussion section. As far as the performance of our standard run is concerned (including spatial distribution of ice sheets) it is discussed in detail in Ganopolski et al. (2010).

***Page 11, Line 14-16 and Conclusions***: *The authors state that the American melt depends largely on the snow melt factor, which can be attributed to the effect of dust deposition. I think the authors need to clarify how dust deposition and snow age interplay in the model. Is the albedo change a linear function of the snow age/dust or do other factors play in? What is the relationship between snow age (simply changes of snow properties) and dust deposition? Could it be other factors that cause these differences?*

The albedo scheme is described in Calov et al (2005). Indeed, albedo depends on snow age, concentration of impurities, and the effect of latter depends on snow age – the older snow is – the large effect of impurities. We now devoted entire new section 5 and two figures (12 and 13) discussing the role of parameterization of snow albedo on simulation of glacial cycle.

***Page 11, Line 25 to 31***: *Fig. 12 needs to be explained better. Please clarify this paragraph. Currently it is hard to follow the reasoning.*

The former Figure 12 was replaced by the new one. Please see our response to the comment above and also to Reviewer#1.

**MINOR ISSUES**

Page 1, Line 2: precessional

This sentence was removed

*Page 2, Line 20: . . . meteorological conditions on high frequency time scales – the difference of the input data between SEB and PDD is not clear.*

It is written in the previous paragraph that PDD "requires information only about surface air temperature" while SEB "requires a complete set of meteorological conditions".

*Page 7 and 8: There is a mismatch between the figure order as mentioned in the text*

*and the actual figure order. Fig. 6 before Fig 4 and 5.*

Now Fig. 6 is not mentioned before Fig. 4 and 5

*Page 7, Line 18 to 19: Remove second 'lie in the range'. Repetition.*

Done

Page 7, Line 23: Remove "range". Repetition.

Done

Page 7, Line 30: "." Before "Thereby".

Corrected

Page 9, Line 25: "we use the latter alphaI value and vary alphaS"

*Several times throughout the text: "vice versa" and not "vice verse"*

Corrected

*Fig. 9 and 11: The authors could consider a more realistic map projection.*

We agree that these figures look a bit odd but for producing figures we used MATLAB which does not contain map projections

*Throughout the text: Please revise for language mistakes.*

We made an extensive revision of the entire manuscript

[revised manuscript text omitted]

---

## Referee Report (RR1)

**Review of Bauer and Ganopolski, 2017, CP**
* * *
This version improved largely from the previous CPD version. However, still many small issues remain. Most importantly, the text should be written more concisely and clearer. I give some suggestions below, but many more remain in the text, and I therefore again highly recommend a thorough English language check (by a professional) before this manuscript is ready for publication.

**Comments:**

Page 1, abstract: delete "It uses the SEB … snow aging"

Page 1, lines 6-7: "The standard version of the CLIMBER-2 model uses the SEB approach and simulates ice volume …"

Page 1, line 8: "Using these simulated temperatures over the last glacial cycle, we calculate the ablation with …"

Page 1, lines 14-15: "off-line simulations: no combination of PDD parameters simulates the ice sheet evolution realistically…"

Page 1, line 22: "These fluctuations result from the interplay of the processes…"

Page 2, lines 5-6 (as commented on before): Is it correct to say that the surface mass balance is the **main** factor affecting the evolution of ice sheets? What about calving, and basal processes?

Page 2, lines 15+: The PDD methods calculates the amount of ablation, the positive part of the surface mass balance, the accumulation, is not included in the PDD calculations. Please rewrite.

Page 3, line 4: "spatially and temporally dependent" on what?

Page 9, line 2: "mismatch" or similar instead of "discontinuity"

Page 9, line 9: so you mean again (3,(9,16)), right? Make that clear.

Page 9, lines 9-16: can be shortened.

Page 9, line 21: delete "appreciably"

Page 9, lines 23-35; change to " accumulation scheme remains…"

Section 4.1: refer to figure 10

Section 4.2 and 4.3: refer to the colors of the lines in Fig 10 when writing about I3/T3/L3 (e.g. I3 and I5, blue line in Fig 10.) to enhance readability.

Page 9, line 31: rewrite "recurrence to Holocene climate characteristics". Remember climate is more than just temperature.

Figure 11 caption: change "window" to "period"

Page 11, line 8: delete "In nature,"

Section 5 comes totally unexpected. I think that it is outside the scope of this study to discuss the different snow parameterizations in such detail, and suggest removing this section and related figures. If decided otherwise, then extensive rewriting is needed in order to explain its importance with respect to the first part of the study.

Also, page 11, line 9: "ablation occurs only in summer at the ice margin" is not true, please rewrite/delete.

Page 12, line 19: rewrite

---

## Author Response (AR2)

I would like to thank both reviewers for their very helpful comments and suggestions. I modified the manuscript following reviewers' suggestions. Since the reviewers expressed opposite opinions concerning the appropriateness of the new section 5, I decided to remove it from the manuscript with the intention to write another paper solely devoted to the effect of dust on snow albedo and ice sheet mass balance. I also significantly revised the discussion.

In the following, I provide specific responses to each of the points raised in reviewers comments.

**Reviewer 1**

*Page 1, abstract: delete "It uses the SEB … snow aging"*

Done

*Page 1, lines 6---7: "The standard version of the CLIMBER---2 model uses the SEB approach and simulates ice volume …"*

Done. The sentence is rewritten accordingly

*Page 1, line 8: "Using these simulated temperatures over the last glacial cycle, we calculate the ablation with …"*

Simulation of ablation with the PDD scheme requires not only air temperature but also snow accumulation rate. This is why, I did not change this sentence.

*Page 1, lines 14---15: "off---line simulations: no combination of PDD parameters simulates the ice sheet evolution realistically…"*

Done

*Page 1, line 22: "These fluctuations result from the interplay of the processes…"*

Done

*Page 2, lines 5---6 (as commented on before): Is it correct to say that the surface mass balance is the main factor affecting the evolution of ice sheets? What about calving, and basal processes?*

(At least in CLIMBER-2) the surface mass balance is the main factor in the sense that changes in surface mass balance are directly controlled by the orbital forcing and thus represent the primary driver of glacial cycles.

*Page 2, lines 15+: The PDD methods calculates the amount of ablation, the positive part of the surface mass balance, the accumulation, is not included in the PDD calculations. Please rewrite.*

Agree. This is now clarified.

*Page 3, line 4: "spatially and temporally dependent" on what?*

The sentence is modified.

*Page 9, line 2: "mismatch" or similar instead of "discontinuity"*

"discontinuity"  is now replaced by "mismatch"

*Page 9, line 9: so you mean again (3,(9,16)), right? Make that clear.*

It is correct in the terms of Figure 6 caption where we use triple index to make this caption shorter. However, in the rest of the text and in other figure captions with use double index for melt factors only (e.g. (9,16)) and stated the sigma value separately.

*Page 9, lines 9---16: can be shortened.*

The paragraph is shortened considerably

*Page 9, line 21: delete "appreciably"*

Done

Page 9, lines 23---35; change to " accumulation scheme remains…"

*Section 4.1: refer to figure 10*

Done

*Section 4.2 and 4.3: refer to the colors of the lines in Fig 10 when writing about*

*I3/T3/L3 (e.g. I3 and I5, blue line in Fig 10.) to enhance readability.*

Done.

*Page 10, line 31: rewrite "recurrence to Holocene climate characteristics". Remember climate is more than just temperature.*

Agree. Now we only discuss the timing of glacial termination. Holocene climate is not mentioned any more.

*Page  11, line 8: delete "In nature,"*

The entire section 5 is removed

*Section 5 comes totally unexpected. I think that it is outside the scope of this study to discuss the different snow parameterizations in such detail, and suggest removing this section and related figures. If decided otherwise, then extensive rewriting is needed in order to explain its importance with respect to the first part of the study.*

The entire section 5 is removed

*Also, page 11, line 9: "ablation occurs only in summer at the ice margin" is not true, please rewrite/delete.*

The entire section 5 is removed

*Page 12, line 19: rewrite*

This sentence is removed

**Reviewer 2**

*Page 4, Line 6: How dependent are SMB and PDD approaches on the 3-day time step?*

3-day time step was chosen to reduce the computational cost of SEMI which otherwise would slowdown CLIMBER-2 considerably. We compared the results for 1-day and 3-day time steps and found rather small differences for the SEB approach. For the PDD approach they are even smaller.

*Fig. 9 and 11: As response to 'a missing mapping tool in Matlab' - there are two ways to plot maps within matlab. One is to use the build in 'mapping toolbox'. For scientific purposes the 'm_map' package, which is freely available at https://www.eoas.ubc.ca/~rich/map.html, is also very useful.*

I am thankful to the reviewer for this information. Unfortunately, the first author Eva Bauer, is not available anymore to work on this manuscript and redoing the graphics would require too much efforts from me.

*Page 6, Line 5: 'Note, that' not 'then'.*

Corrected

*Page 6, Line 32: Double '.'*

Corrected

*Page 7, Line 6 and 7: 'increases' not 'grows'*

Corrected

*Page 10, Line 10: 'warmer' not 'less cold'*

Corrected

*Throughout the manuscript: There are many definite articles missing, specifically in the Discussion section. Please revise the manuscript throughout for linguistic issues.*

The manuscript and has been revised substantially.

[revised manuscript text omitted]